# The extracellular gate shapes the energy profile of an ABC exporter

Cedric A.J. Hutter [1], M. Hadi Timachi[2], Lea M. Hürlimann [1], Iwan Zimmermann [1], Pascal Egloff [1], Hendrik Göddeke[3], Svetlana Kucher[2], Saša Štefanić[4], Mikko Karttunen [5], Lars V. Schäfer [3], Enrica Bordignon[2] & Markus A. Seeger [1]

ABC exporters harness the energy of ATP to pump substrates across membranes. Extracellular gate opening and closure are key steps of the transport cycle, but the underlying mechanism is poorly understood. Here, we generated a synthetic single domain antibody (sybody) that recognizes the heterodimeric ABC exporter TM287/288 exclusively in the presence of ATP, which was essential to solve a 3.2 Å crystal structure of the outward-facing transporter. The sybody binds to an extracellular wing and strongly inhibits ATPase activity by shifting the transporter's conformational equilibrium towards the outward-facing state, as shown by double electron-electron resonance (DEER). Mutations that facilitate extracellular gate opening result in a comparable equilibrium shift and strongly reduce ATPase activity and drug transport. Using the sybody as conformational probe, we demonstrate that efficient extracellular gate closure is required to dissociate the NBD dimer after ATP hydrolysis to reset the transporter back to its inward-facing state.

---

[1] Institute of Medical Microbiology, University of Zurich, Gloriastr. 28/30, 8006 Zurich, Switzerland. [2] Faculty of Chemistry and Biochemistry, Ruhr University Bochum, 44801 Bochum, Germany. [3] Theoretical Chemistry, Faculty of Chemistry and Biochemistry, Ruhr University Bochum, 44801 Bochum, Germany. [4] Institute of Parasitology, University of Zurich, Winterthurerstrasse 266a, 8057 Zurich, Switzerland. [5] Department of Chemistry and Department of Applied Mathematics, The University of Western Ontario, London, ON N6A 3K7, Canada. Correspondence and requests for materials should be addressed to E.B. (email: enrica.bordignon@rub.de) or to M.A.S. (email: m.seeger@imm.uzh.ch)

ABC exporters are versatile membrane proteins found in all phyla of life. Type I exporters are the best studied class of ABC exporters and minimally consist of two transmembrane domains (TMDs) each comprising six transmembrane helices and two nucleotide binding domains (NBDs) that are universally conserved among all ABC transporters. The NBDs undergo large conformational changes in response to ATP binding and hydrolysis, which are transmitted to the TMDs via coupling helices to assume inward-facing (IF), outward-facing (OF), and outward-occluded (Occ) conformations[1]. Alternating access at the TMDs in conjugation with affinity changes towards the transported substrates enable uphill transport across the lipid bilayer[2]. Fully closed NBDs are stabilized by two ATP molecules bound at the dimer interface and coincide with TMDs adopting an OF or Occ state[3,4]. The transition to the IF state requires the NBDs to separate at least to some degree, a process that is initiated by ATP hydrolysis[5].

Many ABC exporters including the entire human ABCC family exhibit asymmetric ATP binding sites, namely a degenerate site that can bind but not hydrolyze ATP and a consensus site that is hydrolysis-competent[6]. Heterodimeric TM287/288 of the thermophilic bacterium *Thermotoga maritima* was the first structurally analyzed example of an ABC exporter with a degenerate site[7,8]. Two closely related IF structures of TM287/288 were solved by X-ray crystallography either containing one AMP-PNP molecule bound to the degenerate site or no nucleotide. In contrast to most other IF structures of ABC exporters, the opened NBDs of TM287/288 are only partially separated due to contacts mediated by the degenerate site D-loop, whereas the consensus site D-loop was found to allosterically couple ATP binding at the degenerate site to ATP hydrolysis at the consensus site[8]. The consensus site features distortions in the Walker B motif, which prevents nucleotide binding in the IF transporter[7]. DEER studies have revealed that TM287/288 exhibits dynamic IF/OF equilibria in the presence of nucleotides and that nucleotide trapping at the consensus site is required to strongly populate the OF state, whereas in the presence of AMP-PNP the transporter predominantly adopts its IF state[9].

Broad distance distributions were found by DEER in the extracellular gate of TM287/288, hinting at conformational flexibility in this external region[9]. Similar observations were reported for ABCB1[10]. Unbiased Molecular Dynamics (MD) simulations of TM287/288 uncovered spontaneous conformational transitions from the IF state via an Occ intermediate to the OF state[11]. Many simulations remained trapped in the Occ state, suggesting that extracellular gate opening represents a major energetic barrier in the conformational cycle. Interestingly, the degree of extracellular gate opening varies greatly among different type I ABC exporters solved in OF states, whereas the gate remains closed in the Occ state[3,4,12]. Hence, events occurring at the extracellular gate likely play a key role in substrate transport and must be allosterically coupled to the catalytic cycle of the NBDs. Nevertheless, the underlying molecular mechanism is unknown.

In this work, we generated single domain antibodies that exclusively bind to OF TM287/288 and thereby inhibit the transport cycle. The binders were instrumental to solve a crystal structure of the transporter in its OF state and were used to probe molecular events at the extracellular gate and their allosteric coupling with the NBDs.

## Results

### Conformational trapping of TM287/288.
Having solved two closely related IF structures of TM287/288, our aim was to obtain an atomic structure of this heterodimeric ABC exporter in its OF state. DEER analyses revealed that TM287/288 carrying the TM288[E517Q] mutation in the Walker B motif of the consensus site (EtoQ mutation) was almost completely trapped in the OF state in the presence of ATP-Mg or ATPγS-Mg[9]. To further decrease the residual ATPase activity of the EtoQ mutant (turnover of 0.02 min$^{-1}$) by a factor of 6.5, we instead introduced the EtoA mutation. In addition, we generated single domain antibodies (nanobodies) that exclusively recognize the OF state of TM287/288. To this end, alpacas were immunized with OF TM287/288 containing a cross-linked tetrahelix bundle motif[13] (see Methods). This approach yielded nanobody Nb_TM#1 binding exclusively to TM287/288 in the presence (but not in the absence) of ATP, as shown by surface plasmon resonance (SPR) (Fig. 1d). However, crystals obtained with Nb_TM#1 did not diffract well enough to build a reliable model. Therefore, we selected synthetic nanobodies (sybodies) against TM287/288 (EtoA) in the presence of ATP-Mg completely in vitro[14]. Thereby, more than ten OF-specific sybodies were generated and sybody Sb_TM#35 was successfully used to solve the OF structure of TM287/288(EtoA) in the presence of ATPγS-Mg at 3.2 Å resolution (Fig. 1a, Supplementary Table 1).

### Structure of TM287/288-sybody complex.
Sybody Sb_TM#35 binds on top of an extracellular wing of TM287/288 (Fig. 1a) and was crucially involved in establishing crystal contacts (Supplementary Fig. 1). Binding is mediated by aromatic residues of all three complementarity determining regions (CDRs) of the sybody, which are wedged between transmembrane helices (TMs) 1 and 2 of TM287 and TMs 5′ and 6′ of TM288 (Fig. 2a). Since Sb_TM#35 only binds in the presence of ATP (Fig. 1d), we hypothesized that it interferes with the catalytic cycle of the transporter. Indeed, the sybody inhibited the ATPase activity of TM287/288 in detergent (IC$_{50}$ of 66.1 nM, Fig. 2b), as well as reconstituted in nanodiscs (Supplementary Fig. 2b). Of note, inhibition was less efficient in nanodiscs, presumably due to impaired epitope accessibility of the sybody in the membrane context.

### Two nanobodies addressing epitopes on the NBDs.
Using the high resolution structure of OF TM287/288 for molecular replacement and as template for model building, we solved two additional low resolution structures (3.5–4.2 Å) of the OF transporter determined in complex with alpaca nanobodies Nb_TM#1 and Nb_TM#2 (Fig. 1b, c). Nb_TM#1 specifically recognizes the OF state and binds to the bottom of the closed NBD dimer, occupying an epitope that is shared between NBD287 and NBD288 (Fig. 1d). Akin to Sb_TM#35, state-specific Nb_TM#1 was found to inhibit the transporter's ATPase activity (Fig. 2b). Nb_TM#2 binds side-ways to NBD288 and exhibits picomolar affinity for the transporter regardless whether ATP is present or not (Fig. 1d, Supplementary Table 2). Nevertheless, this nanobody partially inhibits ATPase activity by around 30% already at the lowest assayed concentration of 20 nM (Fig. 2b). Because the TM287/288 concentration needed to be at least 8 nM to reliably measure ATPase activity, we could not determine the IC$_{50}$ value for Nb_TM#2. A measurement artifact can be excluded, because an unrelated control sybody did not affect the transporter's ATPase activity (Fig. 2b).

### IF to OF transition renders TM287/288 more symmetric.
The OF structure of TM287/288 features fully dimerized NBDs that sandwich two ATPγS-Mg molecules at the degenerate and the consensus site (Fig. 3a, Supplementary Fig. 3). An almost identical structure (RMSD of 0.21 Å) was also obtained in the presence of ATP-Mg (Supplementary Fig. 4c, Supplementary Table 1). In contrast to the NBDs of IF TM287/288, which exhibited pronounced asymmetries between degenerate and

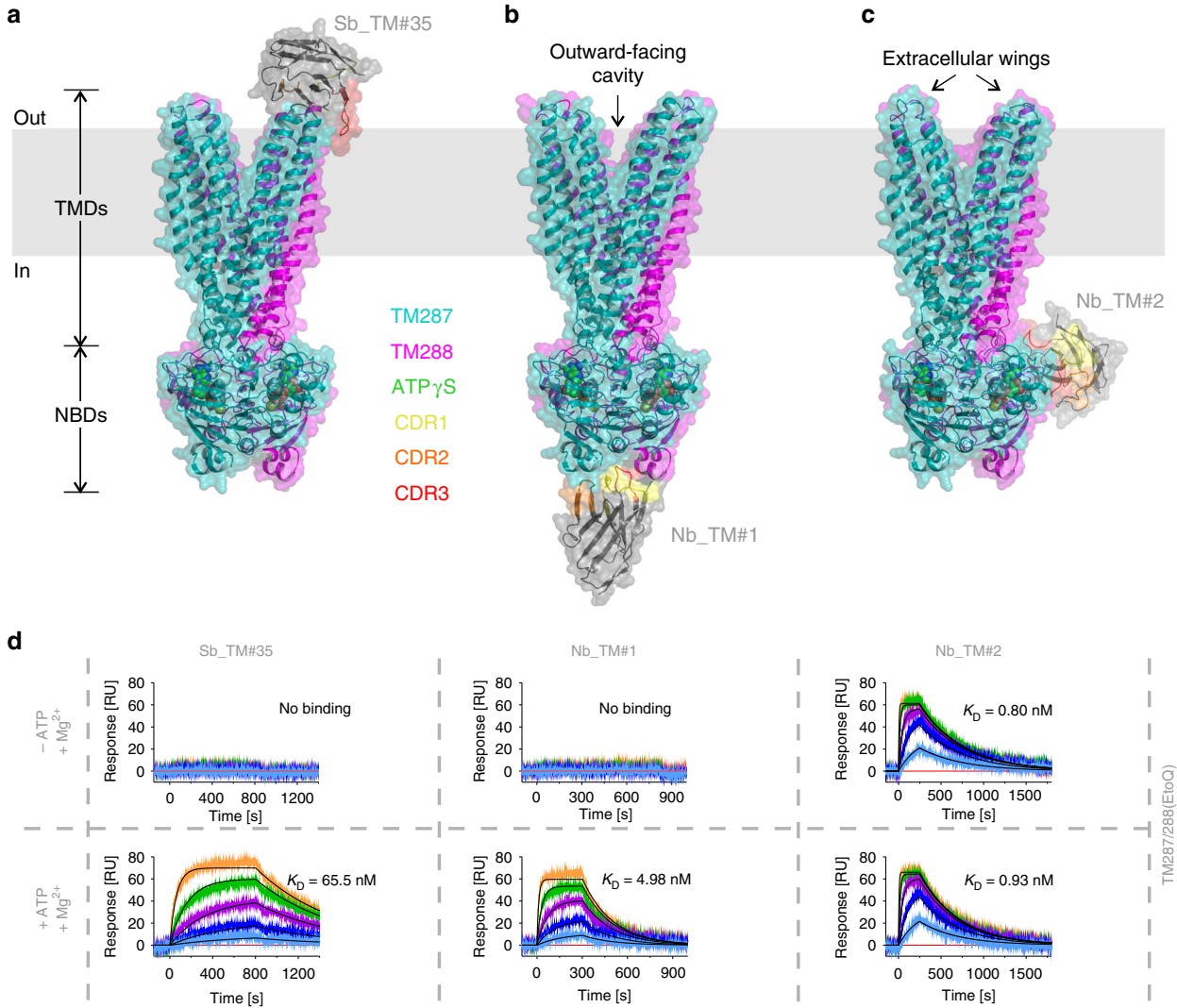

**Fig. 1** Three outward-facing structures of TM287/288 solved in complex with single domain antibody fragments. The transporters are viewed along the membrane plane (indicated as gray rectangle). **a** 3.2 Å structure of TM287/288(EtoA) in complex with ATPγS-Mg and state-specific sybody Sb_TM#35. **b** 3.5 Å structure of TM287/288(2xDtoA/EtoA) in complex with ATPγS-Mg and state-specific nanobody Nb_TM#1. **c** 4.2 Å crystal structure of TM287/288(2xDtoA/EtoA) in complex with ATPγS-Mg and state-unspecific nanobody Nb_TM#2. **d** SPR analyses in the absence (upper panel) and presence (lower panel) of ATP using immobilized TM287/288(EtoQ) as ligand and Sb_TM#35, Nb_TM#1 and Nb_TM#2 as analytes. Injected concentrations of Sb_TM#35: 0, 9, 27, 81, 243, 729 nM; Nb_TM#1: 0, 1, 3, 9, 27, 81 nM; Nb_TM#2: 0, 0.9, 2.7, 8.1, 24.3, 72.9 nM. Kinetic analysis is shown in Supplementary Table 2

consensus site mainly with regard to the D-loops[8], the closed NBD dimer of the OF transporter is more symmetric (Fig. 3a, Supplementary Fig. 3). Further, the distortions found at the catalytic dyad of the consensus site of the IF structure (E517$^{TM288}$ and H548$^{TM288}$) relax during the transition to the OF state and the two key residues adopt a hydrolysis-competent arrangement (Fig. 3b). Interestingly, two tunnels that would allow for release of the cleaved γ-phosphate are present at the consensus site (Fig. 3c). The TMDs consisting of two wings each encompassing six transmembrane helices donated from both protomers are widely opened towards the outside (Supplementary Fig. 4). With an RMSD of 1.73 Å, the structure of TM287/288 most closely resembles the structure of Sav1866. Furthermore, the OF structure is similar to the OF conformation of TM287/288 predicted by MD simulations (Supplementary Fig. 5)[11], although in the MD simulations the protein was embedded in a lipid bilayer instead of the detergent environment used for crystallization. Also the degree of NBD closure and extracellular gate opening is highly similar between TM287/288 and Sav1866 (Supplementary

Fig. 6a). The RMSD between TM287 and TM288 decreases from 2.55 Å to 1.98 Å as the transporter is converted from the IF to the OF conformation, indicating that OF TM287/288 is more symmetric (Supplementary Fig. 6b). Whereas a similar degree of symmetry was observed between the half-transporters of OF ABCB1 (PDB: 6C0V, RMSD of 2.07 Å), the equivalent superimpositions exhibit substantial asymmetries in the OF structure of MRP1 (PDB: 6BHU, RMSD of 4.54 Å), mostly owing to asymmetries in the TMDs (Supplementary Fig. 6b). Extracellular gate opening is less pronounced in MRP1 and even less so in ABCB1, and the gate remains almost completely closed in the outward-occluded structure of McjD[4] (Supplementary Fig. 6b). Hence, structures of OF and Occ ABC exporters show their largest structural variation in the extracellular gates.

**The sybody acts as a molecular clamp.** Interestingly, we did not find steric clashes which would prevent Sb_TM#35 from binding to the IF transporter. Hence, based on structural information

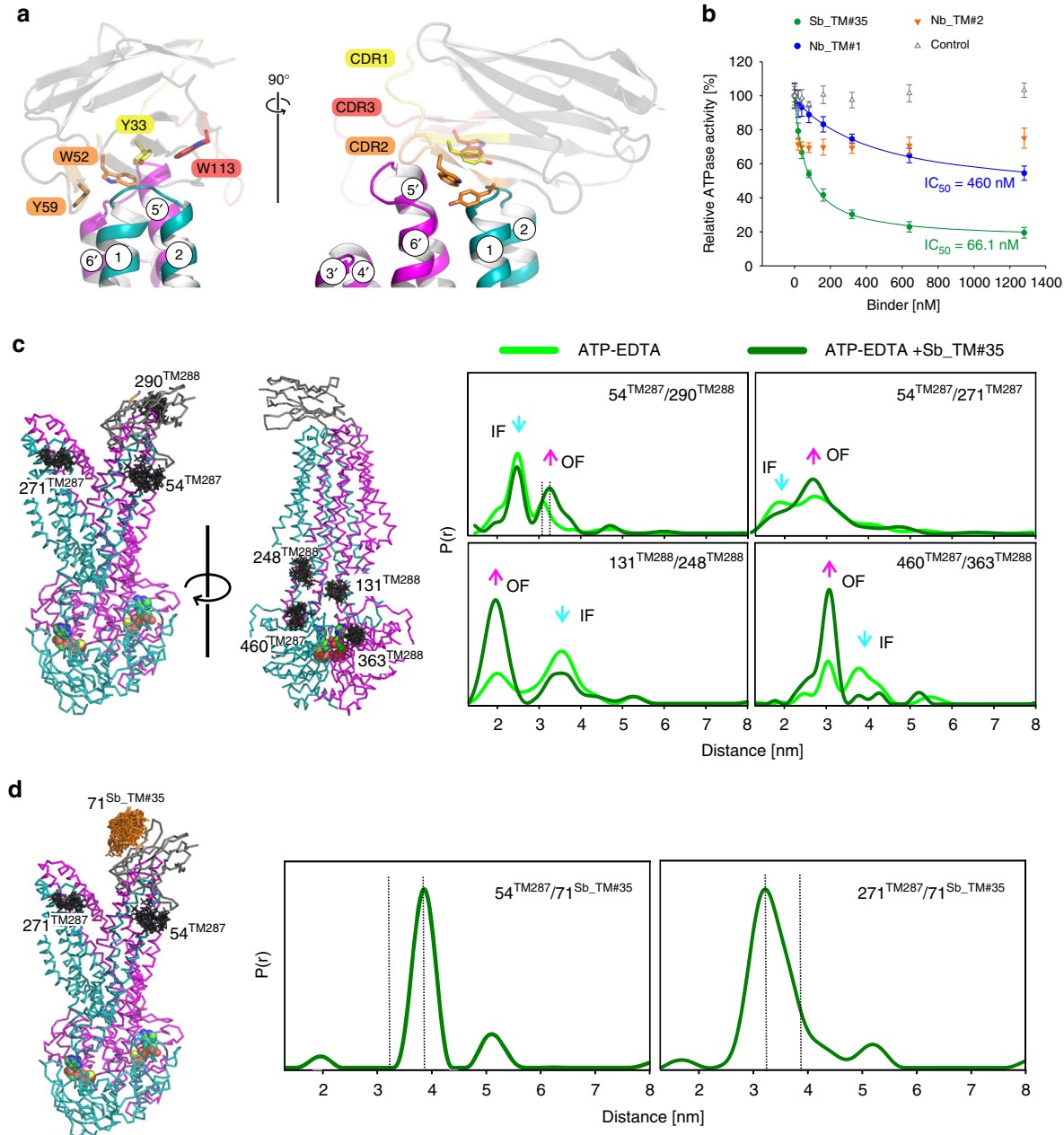

**Fig. 2** The sybody traps TM287/288 in its OF state. **a** Sybody Sb_TM#35 is shown as cartoon in gray with the CDR1, 2 and 3 highlighted in yellow, orange and red, respectively. Four aromatic residues (Y33, W52, Y59, and W113) that wedge between TMs 1 and 2 of TM287 (teal) and TMs 5′ and 6′ of TM288 (magenta) are highlighted as sticks. **b** Inhibition of TM287/288's ATP hydrolysis by Sb_TM#35, Nb_TM#1 and Nb_TM#2. A non-randomized sybody served as control. The data were fitted with a hyperbolic decay function to determine $IC_{50}$ values, as well as residual activities. The error bars are standard deviations of technical triplicates. **c**, **d** DEER analyses of spin-label pairs introduced to probe the extracellular and intracellular TMDs and the NBDs (**c**), as well as sybody binding to the transporter (**d**). DEER traces were recorded in the presence of ATP-EDTA with or without unlabeled Sb_TM#35 (**c**) or in the presence of ATP-EDTA and spin-labeled Sb_TM#35 (**d**). The graphs show experimental distance distributions, and vertical dotted lines shown in **c** highlight changes in the mean distances

alone we could not explain why the sybody inhibits ATPase activity. Therefore, we used DEER spectroscopy to unravel the sybody's impact on the conformational cycle.

The sybody was found to shift the transporter's equilibrium towards the OF state, as measured in the presence of ATP-EDTA (arrows in Fig. 2c). Pronounced effects were observed in the extracellular region ($54^{TM287}/290^{TM288}$ and $54^{TM287}/271^{TM287}$), but also when probing distances at the intracellular region of the TMDs ($131^{TM288}/248^{TM288}$) and at the NBDs ($460^{TM287}/$

$363^{TM288}$) (Fig. 2c and Supplementary Fig. 7). Further, we observed a distance increase between two spin labels positioned in the wing underneath the sybody ($54^{TM287}/290^{TM288}$) as a result of sybody binding (dotted vertical lines in Fig. 2c). This suggests that the sybody acts as a wedge at the opened extracellular wing. As expected from the lack of sybody binding to the IF state, we observed negligible effects on the interspin distances when TM287/288 was incubated with the sybody in the absence of nucleotides (apo state) (Supplementary Fig. 7).

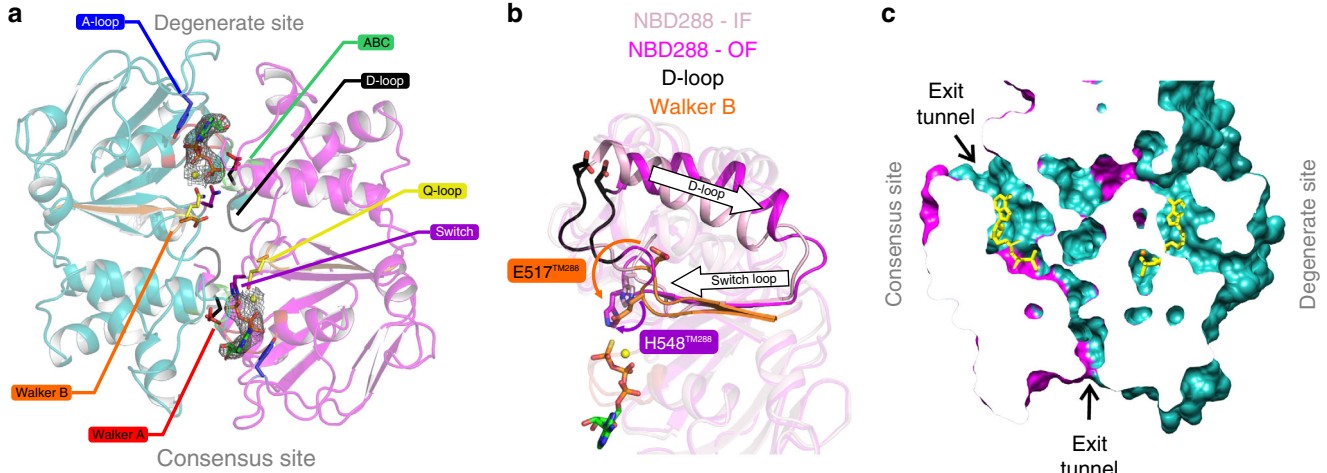

**Fig. 3** Structural analysis of the closed NBD dimer. **a** The fully closed NBD dimer (NBD287 in teal and NBD288 in magenta) sandwiches two ATPγS-Mg molecules (shown as sticks with corresponding electron density) between Walker A motif (red) and the opposite ABC signature motif (green) at the degenerate and the consensus site in a highly symmetric manner. Residues involved in ATP binding and hydrolysis are shown as sticks. **b** Superimposition of the consensus ATP binding site of the previously solved IF structure (PDB: 4Q4A, light pink) and the OF structure (magenta). Distortions of the catalytic dyad (E517$^{TM288}$ and H548$^{TM288}$) are relaxed during NBD closure to adopt a hydrolysis-competent arrangement. The side chain of E517$^{TM288}$ was modeled into the TM287/288(EtoA) structure. **c** Slice-cut through the two nucleotide binding sites reveals two possible P$_i$ exit tunnels at the consensus site which are not present at the degenerate site. ATPγS (partially clipped) is shown as yellow sticks

To investigate the positioning of the bound sybody relative to the opposite wing, we then focused on the distance between the sybody labeled at position 71 and spin labels introduced either at 54$^{TM287}$ (the sybody-binding wing) or 271$^{TM287}$ (opposite wing) of the transporter (Fig. 2d and Supplementary Fig. 8). The main distance peak corresponding to dipolar coupling between 71$^{Sb\_TM\#35}$ and 54$^{TM287}$ was very sharp and centered at 3.8 nm, while it was somewhat broader and centered at 3.2 nm between 71$^{Sb\_TM\#35}$ and 271$^{TM287}$ placed on the opposite wing. Both distances were visible only in the presence of ATP, and were in close agreement with the simulations based on the OF structure (Supplementary Fig. 8). Both traces also contained a distance peak at around 5.2 nm corresponding to a residual fraction of sybody dimers in solution. In conclusion, the sybody acts as a molecular clamp that keeps the extracellular gate open.

**Conserved aspartates seal the extracellular gate.** Having shown that the sybody traps the transporter in a fully opened state, we reasoned that mutations facilitating extracellular gate opening would have a similar impact on the transporter's energy landscape. In IF TM287/288, D41$^{TM287}$ and D65$^{TM288}$ placed in TM1 of the respective half-transporter establish hydrogen bonds with backbone amides of the opposite wing (Fig. 4a). Of note, these aspartates are conserved in bacterial ABC exporters (Fig. 4b), but not in eukaryotic members of the family. When the aspartates were substituted with alanines, the ATPase activity of TM287/288 decreased around three-fold for the single mutants and around 10-fold for the double mutant (henceforth called 2xDtoA mutant) (Fig. 4c).

Using again ATP-EDTA to induce the IF to OF transition, DEER analyses revealed an equilibrium shift towards the OF state in the 2xDtoA mutant for all spin-labeled pairs (Fig. 4f and Supplementary Fig. 9). The equilibrium shift was similar to that induced by Sb\_TM\#35. Hence, the aspartates at the extracellular gate constitute an energy barrier that needs to be overcome to switch to the OF state and influence the ATPase cycle in a long-ranged allosteric coupling connecting the extracellular gate with the NBDs.

To probe the atomic details of the conformational dynamics underlying the IF–OF transition, we performed MD simulations of TM287/288 in a POPC lipid bilayer starting from the IF crystal structure (PDB: 4Q4A), after docking a second ATP-Mg molecule into the consensus site[11] and introducing the 2xDtoA mutations in the extracellular gate. As in our previous MD simulations of wild-type TM287/288[11], we observed spontaneous large-scale conformational transitions from the IF conformation via an Occ state to an OF conformation; this complete transition was observed in 3 out of 20 independent 500 ns simulations (Supplementary Fig. 10). Despite the limited statistics, the transition appears to be slightly more frequent than for the wild-type (6 out of 100 simulations[11]), in agreement with our experimental results and the notion that the polar contacts of the two aspartate residues increase the energy barrier of extracellular gate opening. Although MD simulations and experimental data are in agreement, we cannot exclude different results if these rather long simulations were conducted in a lipid bilayer containing other lipids such as for example POPE[15]. To assess the stability of the OF structure reported in this work, 20 independent 400 ns simulations were carried out for both the wild-type and the 2xDtoA mutant. Although the sybody is not present in the simulations, the OF conformation with two bound ATP-Mg molecules is very stable and merely fluctuates around the X-ray structure (Supplementary Fig. 11). Additional control simulations in a POPE (instead of POPC) bilayer confirmed this result (Supplementary Fig. 11), rendering it unlikely that lipid composition (in terms of PC vs. PE head groups) has a large effect on the ATP-Mg-bound OF structure.

Next, we introduced the 2xDtoA mutations into the hetero-dimeric ABC exporter EfrEF of *Enterococcus faecalis* (Fig. 4b)[16]. Ethidium-stimulated ATPase activity profiles of membrane reconstituted EfrEF were found to be strongly affected by the single DtoA mutations, and the ATPase activity of EfrEF containing the 2xDtoA mutations could no longer be stimulated by the drug (Fig. 4d). Supporting this notion, the TM287/288(2xDtoA) mutant reconstituted in nanodiscs exhibited strongly diminished drug stimulation by Hoechst 33342

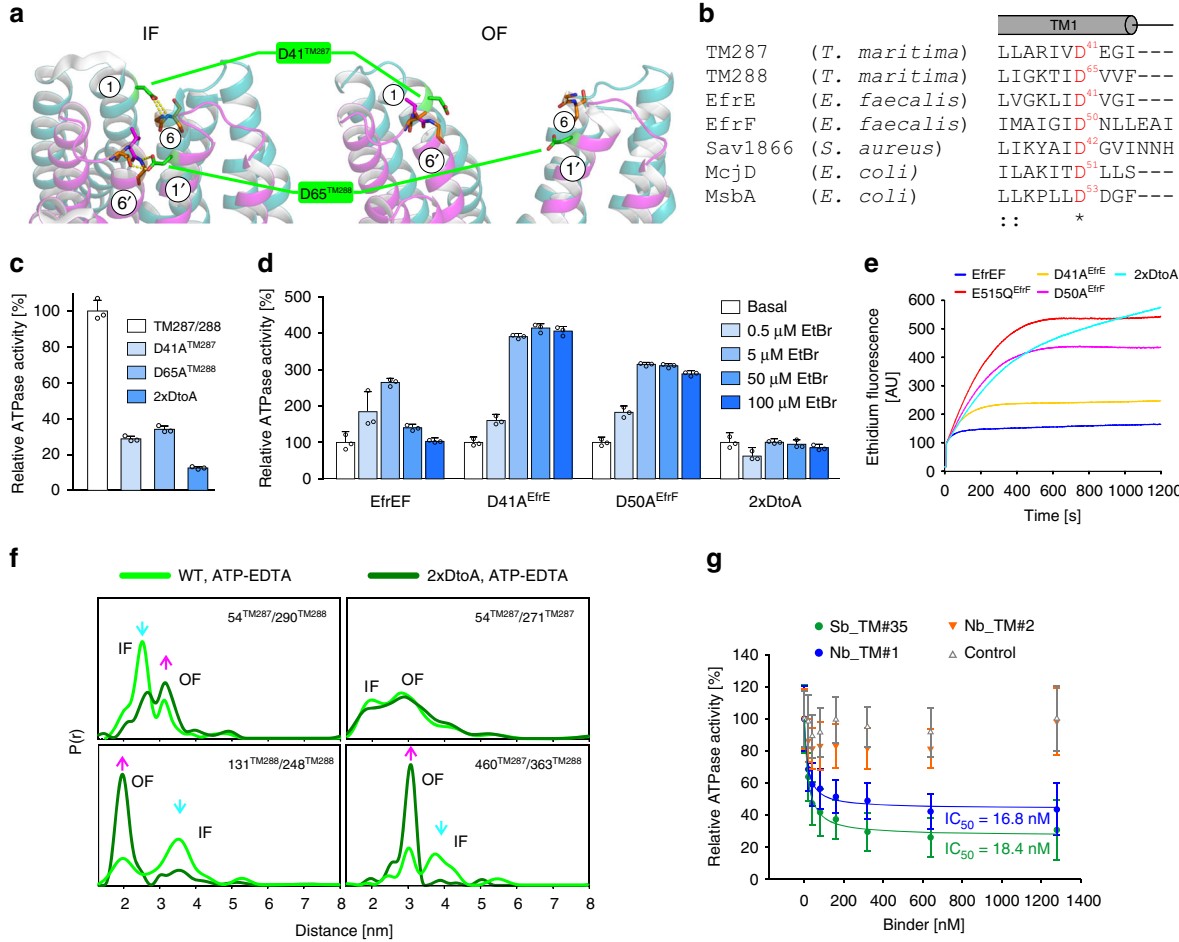

**Fig. 4** The extracellular gate is sealed by two conserved aspartates. **a** Structure of TM287/288's extracellular gate in the IF (left, PDB: 4Q4H) and OF (right) state shown as cartoon. D41$^{TM287}$and D65$^{TM288}$ are shown as sticks and establish hydrogen bonds (dashed yellow lines) with the peptide backbone (shown as sticks) of neighboring helices TM6 and TM6' that are broken during IF−OF transition. **b** Sequence alignment of bacterial ABC exporters in the region containing the conserved extracellular gate aspartates. **c** ATPase activities of single mutants D41A$^{TM287}$ and D65A$^{TM288}$ and the corresponding double mutant (2xDtoA) relative to wild-type TM287/288 determined in detergent. **d** Drug stimulated ATPase activities of wild-type EfrEF, the single mutants D41A$^{EfrE}$ and D50A$^{EfrF}$ and the corresponding double mutant (2xDtoA) reconstituted into proteoliposomes determined in the absence (basal activity) or in the presence of ethidium at the concentrations indicated. Data were normalized to the basal ATPase activity of the respective mutant. The error bars are standard deviations of technical triplicates. **e** Ethidium accumulation of *Lactococcus lactis* cells expressing wild-type EfrEF, the inactive Walker B mutant E515Q$^{EfrF}$ or the extracellular gate mutants D41A$^{EfrE}$ and D50A$^{EfrF}$ or the corresponding double mutant (2xDtoA). **f** DEER analyses probing the extracellular and intracellular TMDs and the NBDs (same positions as in Fig. 2c). DEER traces were recorded in the presence of ATP-EDTA for the wild-type transporter and for TM287/288(2xDtoA). **g** Relative ATPase activities of the 2xDtoA mutant in the presence of increasing concentrations of Sb_TM#35, Nb_TM#1 and Nb_TM#2. A non-randomized sybody served as control. The data were fitted with a hyperbolic decay function to determine IC$_{50}$ values as well as residual activities. The error bars are standard deviations of technical triplicates

(Supplementary Fig. 2a). Next, we expressed EfrEF wild-type and DtoA mutants in *Lactococcus lactis* and monitored ethidium uptake by fluorescence measurements (Fig. 4e). Wild-type EfrEF efficiently expels ethidium from the cell, resulting in a slow increase of ethidium accumulation that reaches a low steady-state level. EfrEF containing the EtoQ mutation in the NBDs served as negative control exhibiting high ethidium accumulation levels[17]. The single DtoA mutants D41A$^{EfrE}$ and D50A$^{EfrF}$ partially lost their capability of ethidium efflux. Interestingly, the accumulation curve of the 2xDtoA mutant does not reach a steady-state level within the time frame of the experiment. This observation suggests a transporter defect resulting in passive influx of ethidium into the cell mediated by EfrEF carrying the 2xDtoA mutations. In conclusion, the extracellular aspartates are important gate-keeper residues that are allosterically coupled to the NBDs and are required for substrate transport.

**Extracellular gate mutant and sybody are synergistic.** Because both sybody binding and weakening of the extracellular gate shifted the conformational equilibrium towards the OF state, we reasoned that these effects are additive. Indeed, with an IC$_{50}$ of 18.4 nM (Fig. 4g), inhibition of TM287/288 carrying the 2xDtoA mutations by Sb_TM#35 was found to be more pronounced than the inhibition of the wild-type transporter (IC$_{50}$ = 66.1 nM) (Fig. 2b). In further agreement, the affinity of Sb_TM#35 towards the 2xDtoA mutant ($K_D$ = 14 nM) was around eight times higher than towards the wild-type transporter ($K_D$ = 110 nM) (Supplementary Fig. 12, Supplementary Table 2). Affinity was as well increased upon introduction of the EtoA or EtoQ mutation into the consensus site of the transporter ($K_D$ = 66 nM) which as well exhibit a conformational equilibrium shift towards the OF state[9] and was highest for the combined triple mutant (2xDtoA/EtoA) ($K_D$ = 8 nM). An analogous pattern was observed for Nb_TM#1,

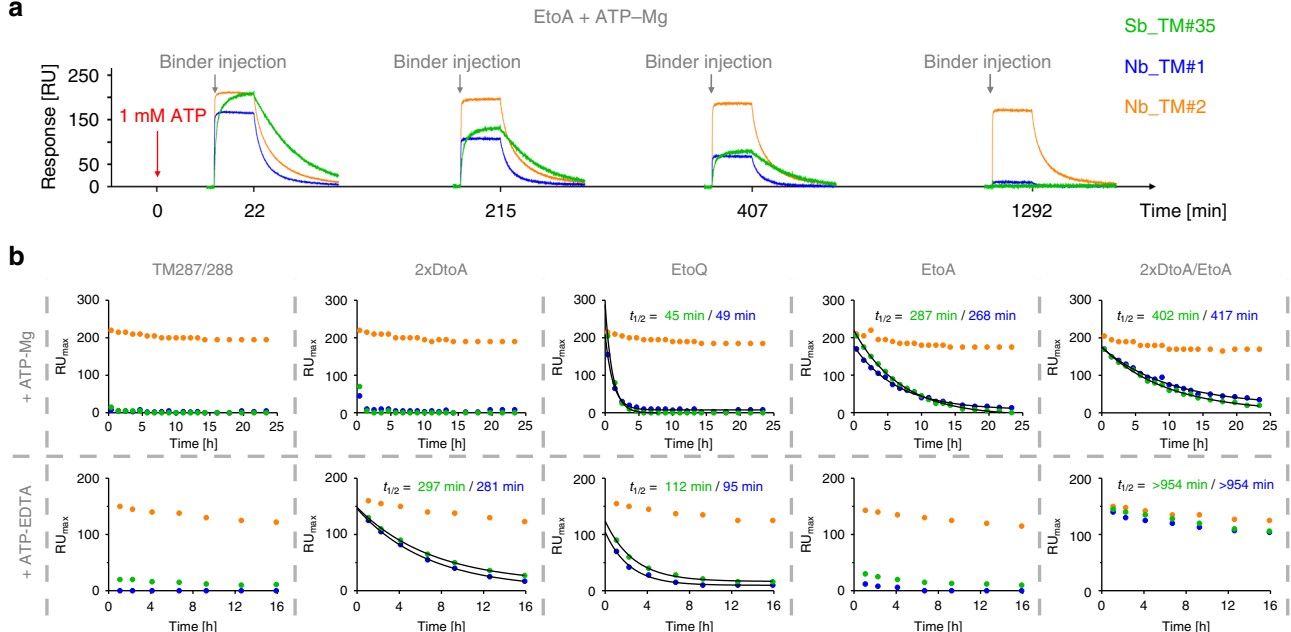

**Fig. 5** Probing the OF–IF transition using state-specific binders. **a** Exemplary raw data of SPR traces based on which conformational probing was detected. At time point zero, immobilized TM287/288(EtoA) was charged with 1 mM ATP (red arrow) and binders were injected (gray arrows) at saturating concentrations (Sb_TM#35, 1 μM; Nb_TM#1, 500 nM; Nb_TM#2, 100 nM) to obtain maximal response unit values ($RU_{max}$) at the indicated time points. **b** $RU_{max}$ values for wild-type and mutant TM287/288 were obtained as shown in **a** by charging the transporter with 1 mM ATP either in presence of $Mg^{2+}$ (upper panel) or EDTA (lower panel). For state-specific binders Sb_TM#35 and Nb_TM#1, data were fitted using a one phase decay function to determine half-life values ($t_{1/2}$) of the OF state. State-unspecific Nb_TM#2 was used as a control

which binds to the closed NBD dimer. The $IC_{50}$ was substantially smaller when probing the 2xDtoA mutant ($K_D = 16.8$ nM) compared to wild-type TM287/288 ($K_D = 460$ nM) (Fig. 2b, Fig. 4g). This difference was again reflected by an affinity increase for the 2xDtoA mutant (37 nM) vs. the wild-type transporter ($K_D = 184$ nM) and the highest affinity was observed for the triple mutant ($K_D = 5$ nM) (Supplementary Fig. 12, Supplementary Table 2). In conclusion, Sb_TM#35 and Nb_TM#1 bind to the opposite ends of the transporter but nevertheless exhibit a highly similar biophysical behavior of trapping the OF transporter.

**Probing the OF–IF conversion by state-specific binders.** We finally tested whether the state-specific nanobodies could be used as probes in SPR to investigate the OF–IF transition of TM287/288. When immobilized TM287/288(EtoA) was charged with ATP-Mg and subsequently washed with buffer devoid of nucleotides, the maximal SPR binding signal for the OF state-specific binders Sb_TM#35 and Nb_TM#1 slowly decreased over a time window of several hours (Fig. 5a). Immobilized TM287/288 was stable within this time frame, because the maximal binding signal for the state-unspecific nanobody Nb_TM#2 only slightly decreased (Fig. 5a). Subsequently, we interrogated the OF–IF conversion using either ATP-Mg (hydrolyzing conditions) or ATP-EDTA (ATP binding without hydrolysis).

In the case of wild-type TM287/288 loaded with ATP-Mg or ATP-EDTA, the state-specific binders were unable to recognize the transporter within the time resolution of the experiment, showing rapid conversion to the IF state. In contrast, the OF state was long-lived when ATP-Mg was occluded by the EtoQ or the EtoA mutant. The lifetime of the OF state as probed by the state-specific binders Sb_TM#35 or Nb_TM#1 was highly similar ($t_{1/2}$ of 45 or 49 min for the EtoQ mutant and 287 or 268 min for the EtoA mutant, respectively). Strikingly, these values are in close agreement with the half-life of ATP hydrolyzed by these mutants,

namely 32 min (EtoQ mutant) and 205 min (EtoA mutant) at 25 °C. For the same mutants, the situation was inversed for ATP-EDTA. The EtoA mutant readily converted to the IF state akin to the wild-type transporter, whereas the OF state was very stable for the EtoQ mutant ($t_{1/2} = 112$ or 95 min, respectively). In this case, ATP cannot be hydrolyzed and we investigate NBD dissociation with bound ATP (but lacking the coordination by $Mg^{2+}$). The glutamine in the Walker B motif appears to stabilize ATP binding in the NBD sandwich dimer, whereas the canonical glutamate or an alanine at the same position promotes fast NBD dissociation. When the 2xDtoA mutations were introduced, the OF state was very long-lived in case of ATP-EDTA ($t_{1/2} = 297$ or 281 min, respectively). This suggests that weakening of the extracellular gate by the 2xDtoA mutations strongly impedes NBD dissociation, whereas NBD dissociation of the wild-type transporter is very fast under these experimental conditions. NBD opening is further slowed down if the 2xDtoA mutations are combined with the EtoA mutation for both ATP-Mg and ATP-EDTA.

From this dataset one can draw two major conclusions. First, ATP hydrolysis weakens the NBD dimer and is required to reset the transporter to the IF state under physiological conditions where ATP and $Mg^{2+}$ are always present. And second, strong contacts at the extracellular gate are mandatory to exert a mechanical force onto the NBDs to facilitate fast NBD opening in order to reset the transport back to its IF state.

## Discussion

In this work we unleashed the power of state-specific single domain antibodies obtained from alpacas and entirely in vitro from synthetic libraries to investigate a membrane transporter at the structural and functional level. The strategy of generating state-specific binders against type I ABC exporters has a long history going back to the 90's of the last century, when the state-specific ABCB1 antibody UIC2 was identified[18]. A recent

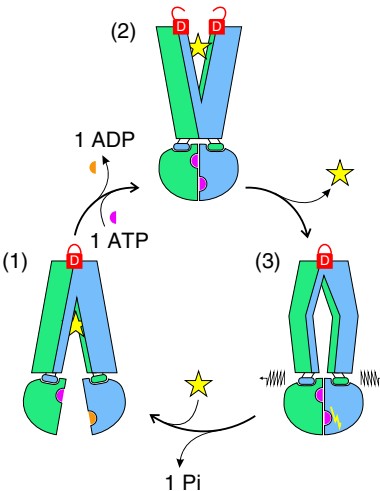

**Fig. 6** Role of the extracellular gate in the transport cycle of TM287/288. Substrate (yellow star) binds to the IF transporter (1) with high affinity, while the extracellular gate is sealed by two aspartates (closed D-lock). Binding and occlusion of two ATP-Mg molecules at the NBD interface drives the transition to the OF state (2). The extracellular gate opens (open D-lock) and substrate is released. The extracellular gate of the empty outward-oriented cavity closes (3) and thereby may trigger ATP hydrolysis at the consensus site. The mechanical force of the firmly sealed extracellular gate (closed D-lock) is required to dissociate the NBDs after ATP hydrolysis in order to reset the transporter to its IF state

cryo-EM structure of ABCB1 in complex with UIC2 revealed that the antibody clamps the extracellular loops together, thereby preventing extracellular gate opening[19]. Molecular clamping of the closed extracellular gate was also achieved by a cyclic peptide raised against CmABCB1[20]. Further examples of binders that prevent the IF–OF conversion are nanobodies raised against ABCB1 and PglK, which both sterically clash with NBD closure[21,22]. In contrast, our binders are specific for the OF state and consequently impede the OF–IF conversion.

Sybody binding to an extracellular wing of TM287/288 was essential to solve the OF structure. Both the degenerate and the consensus ATP binding site are fully closed and highly symmetric, but only the consensus site bears the catalytic dyad positioned to catalyze ATP hydrolysis[23]. This suggests that ATP hydrolysis of only one nucleotide is sufficient to initiate dissociation of the NBDs. In further support of this view, the closed NBD dimers features two possible $P_i$ exit tunnels at the consensus site.

A comparison with other OF and Occ transporter structures revealed major conformational heterogeneity in the degree of extracellular gate opening, which has been discussed to play a potential role in squeezing out substrates or to prevent rebinding of substrates[3,4,10,12,24]. In this study, we uncover cross-talk between the extracellular gate and the ATPase cycle, a connection that has to the best of our knowledge not yet been investigated at the molecular level (Fig. 6). A sybody stabilizing the opened extracellular wing or mutations weakening the extracellular gate both shifted the conformational equilibrium towards the OF state. Previous experimental and computational studies have uncovered that NBD closure precedes extracellular gate opening during the IF–OF transition[11,25], and DEER analyses have revealed that extracellular gate opening can be partial while NBD closure is complete[9,10,26]. Hence, there seems to be an inbuilt mechanical principle that the extracellular gate is energetically costly to open.

Conversely, using our state-specific nanobodies as conformational probes we were able to show that extracellular gate closure is coupled to the dissociation of the closed NBD dimer after ATP hydrolysis. Our experiments on EfrEF demonstrated that a firmly sealed extracellular gate is in fact crucial for transporter function. Further, the 2xDtoA mutant had a strongly reduced ATPase activity and lost its capacity to be stimulated by drugs. This suggests that extracellular gate closure has become the rate-limiting step of the catalytic cycle of the 2xDtoA mutant. Hence, in this mutant the IF–OF conversion is no longer rate-limiting and consequently cannot be stimulated by drug binding to the inward-oriented high-affinity site (Fig. 6).

The ATP-bound OF state with completely closed NBDs has been referred to as the high-energy state of the transport cycle and in some instances it was proposed that ATP hydrolysis is needed to populate the OF state at all[10,26]. In contrast, we and others have previously stipulated that ATP binding alone is sufficient for IF–OF conversion and substrate release[9,11,27], in agreement with the ATP-switch model[28]. It should be noted that while the opened extracellular gate indeed represents a high-energy state, the opposite is in fact true for the closed NBD dimer. It adopts a low-energy state and a large energy input is required to dissociate the dimer[29]. This is certainly achieved in part by the hydrolysis of ATP[30,31]. Importantly, our results suggest that NBD dissociation also involves a mechanical component mediated by extracellular gate closure. A further possibility is the triggering of ATP hydrolysis as a result of extracellular gate closure. Although speculative and neither directly supported nor excluded by our data, such a mechanism would assure that the transporter only reverts to the IF state after substrate release.

Why and when ATP hydrolysis is required to achieve active transport is a recurrent debate in the ABC transporter field. Our data presented here and in previous studies[9,11] clearly suggest that ATP binding alone (in case of ATP-EDTA when ATP hydrolysis cannot occur) is sufficient for the IF–OF conversion and presumably the active transport of one substrate molecule. Directionality of transport is then achieved by an affinity switch of the substrate binding site, which inevitably undergoes drastic rearrangements as the TMDs switch from an IF to an OF conformation[12]. Nevertheless, it is unlikely that an ABC exporter can operate efficiently by binding and dissociation of ATP alone, because energy input is lacking and the molecular events would be driven by slow stochastic Brownian motions alone. To strongly populate the OF state under physiological conditions, ATP-Mg needs to be occluded at the consensus site of the closed NBD dimer[32], a state that can be efficiently mimicked by the Walker B EtoQ or EtoA mutation[33]. As we show here with our binder probes, ATP occlusion firmly traps the transporter in the OF state and prevents transporter cycling. Hence, ATP hydrolysis appears to be strictly required to initiate dissociation of the closed NBD dimer. Once ATP is hydrolyzed, the force exerted by the closed extracellular gate facilitates NBD dissociation. In summary, our results support the notion that ATP hydrolysis is required to drive the transport cycle at the resetting step from the OF to the IF state.

We hope that our results provide a mechanistic framework to further study the functional role of the extracellular gate of type I ABC exporters and to investigate the molecular underpinning of disease-causing mutations found in the extracellular region of medically important ABC exporters such as MRP1 and CFTR[34,35].

## Methods
**Expression and purification**. The genes encoding the heterodimeric ABC transporter TM287/288 were amplified and cloned into pINIT_cat (addgene: Plasmid #46858)[7].The genes were subcloned into pBAD expression vectors by FX cloning;[36]

for crystallization and biochemical analyses into the pBXNH3L expression vector or into the pBXNH3LCA expression vector for the production of biotinylated TM287/288 variants[14].

Freshly transformed MC1061 E. coli cells were grown in Terrific Broth (TB) medium, supplemented with 100 µg/ml ampicillin, to an $OD_{600}$ of 1.0–1.5 at 37 °C and expression was induced by the addition of 0.0017 % (w/v) L-arabinose for 5 h at 30 °C. Cells were harvested and cellular membranes prepared in 20 mM Tris-HCl pH 7.5, 200 mM NaCl and 10 % (v/v) glycerol. Membranes were solubilized by the addition of 1 % (w/v) n-dodecyl-β-D-maltoside (β-DDM, Glycon) for 2 h at 4 °C. Solubilized membranes were supplemented with 20 mM imidazole and loaded on a Ni-NTA Superflow (Qiagen) gravity flow column at 4 °C, washed with 20 column volumes 50 mM imidazole pH 7.5, 200 mM NaCl, 10 % (v/v) glycerol and 0.03 % (w/v) β-DDM or 0.3 % (w/v) n-decyl-β-D-maltoside (β-DM, Glycon) and eluted with 4 column volumes 200 mM imidazole pH 7.5, 200 mM NaCl, 10 % (v/v) glycerol and 0.03 % (w/v) β-DDM or 0.3 % (w/v) β-DM, followed by desalting using a PD-10 column (GE Healthcare, 17–0851–1) equilibrated with 20 mM Tris-HCl pH7.5, 150 mM NaCl and 0.03 % (w/v) β-DDM or 0.3 % (w/v) β-DM. The $His_{10}$-tag was cleaved off overnight at 4 °C using 3 C protease (1:10 w/w), followed by reloading on a Ni-NTA gravity flow column equilibrated with 20 mM Tris-HCl pH 7.5, 150 mM NaCl, 40 mM imidazole and 0.03 % (w/v) β-DDM or 0.3 % (w/v) β-DM at 4 °C. Processed TM287/288 was polished by size-exclusion chromatography using a Superdex 200 Increase 10/300 GL (GE Healthcare) column equilibrated in 20 mM Tris-HCl pH 7.5, 150 mM NaCl and 0.03 % (w/v) β-DDM or 0.3 % (w/v) β-DM at 4 °C and concentrated to the desired concentration with an Amicon Ultra-4 concentrator unit with a molecular weight cut-off (MWCO) of 50 kDa. Purified TM287/288 variants were used immediately or flash-frozen in liquid nitrogen and stored at −80 °C.

Avi-tagged TM287/288 variants were enzymatically biotinylated using BirA during 3 C cleavage in 20 mM imidazole pH 7.5, 200 mM NaCl, 10 % (v/v) glycerol, 10 mM magnesium acetate and 0.03 % (w/v) β-DDM, and two-fold molar excess of biotin.

EfrEF was amplified from the genomic DNA of Enterococcus faecalis V583, cloned into the shuttle vector pREXNH3 via FX cloning[36] and subcloned into the expression vector pNZ8048NH3 via vector backbone exchange cloning (VBEx)[37]. Transformed L. lactis NZ9000 ΔlmrAΔlmrCD cells[38] were grown in M17 medium supplemented with 0.5% glucose and 5 µg/ml chloramphenicol to an $OD_{600}$ of 1 at 30 °C and the expression was induced by adding a nisin-containing culture supernatant of L. lactis NZ9700 for 4 h (1:5000 [v/v]). Cells were harvested and membranes were prepared in 20 mM Tris-HCl pH 7.5, 200 mM NaCl and 10 % (v/v) glycerol[16]. EfrEF was purified using β-DDM in the same way as TM287/288 (see above).

Nanobodies/sybodies were either expressed from pSb_init (addgene: #110100) for biochemical experiments or subcloned into pBXNPHM3 (addgene: #110099) by FX cloning for the production of tag-free nanobodies/sybodies for crystallization and DEER experiments[14,36]. Purified nanobodies and sybodies were stored at −80 °C.

**Mutagenesis.** To weaken the extracellular gate, two conserved aspartates were replaced by alanines in various TM287/288 variants resulting in TM287/288 (2xDtoA). $D41A^{TM287}$ was introduced using the primers TM287_D41A_FW (GGC ACG TAT TGT CGC CGA AGG AAT CG C) and TM287_D41A_RV (GCG ATT CCT TCG GCG ACA ATA CGT GCC). $D65A^{TM288}$ was introduced using the primers TM288_D65A_FW (CAT AGG AAA AAC GAT CGC TGT TGT CTT CG) and TM288_D65A_RV (CGA AGA CAA CAG CGA TCG TTT TTC CTA TG). In order to render TM287/288 catalytically inactive, $E517A^{TM288}$ was introduced in wild-type TM287/288 and TM287/288(2xDtoA) using the primers TM288_E517A_FW (CCT GAT ACT GGA CGC AGC CAC CAG CAA C) and TM288_E517A_RV (GTT GCT GGT GGC TGC GTC CAG TAT CAG G). Mutation $D41A^{EfrE}$ was introduced using the primers EfrE_D41A_FW (CAA GTT GAT TGC TGT GGG CAT CG) and EfrE_D41A_RV (CGA TGC CCA CAG CAA TCA ACT TG). $D50A^{EfrF}$ was generated using the primers EfrF_D50A_FW (CAA TCG GGA TTG CTA ACC TCT TAG AAG C) and EfrF_D50A_RV (GCT TCT AAG AGG TTA GCA ATC CCG ATT G). In order to cross-link the transporter at the tetrahelix bundle in the OF state, $L200C^{TM287}$ was introduced in cys-less TM287/288 (described in ref. [7]) using the primers TM287_L200C_FW (GAG AAA ATC TCT GCG GTG TCA GGG TAG TGA G) and TM287_L200C_RV (CTC ACT ACC CTG ACA CCG CAG AGA TTT TCT C) and combined with the $S224C^{TM288}$ mutation using the primers TM288_S224C_FW (CAT AGA AGA AGA CAT CTG CGG CCT CAC TGT G) and TM288_S224C_RV (CAC AGT GAG GCC GCA GAT GTC TTC TTC TAT G).

For spin labeling of the sybody, S71C was introduced in Sb_TM#35 using the primers Sb_TM#35_S71C_FW (CAC GGT GTG CCT GGA CAA CG) and Sb_TM#35_S71C_RV (CGT TGT CCA GGC ACA CCG TG).

For spin labeling of TM287/288, three new cysteines were introduced in cys-less TM287/288 (called wild-type TM287/288 for simplicity). $S271C^{TM287}$ was generated using the primers TM287_S271C_FW (CAG ATG GAG ATA GGA TGC ATC ATG GCA TAC) and TM287_S271C_RV (GTA TGC CAT GAT GCA TCC TAT CTC CAT CTG) as a single mutant or in combination with $K54C^{TM287}$, which was generated using the primers TM287_K54C_FW (CTT TTC TCT GGT TTT GTG TAC AGG GAT CCT CAT G) and TM287_K54C_RV (CAT GAG GAT

CCC TGT ACA CAA AAC CAG AGA AAA G). $K54C^{TM287}$ was also prepared as a single mutant or in combination with $I290C^{TM288}$ introduced with the primers TM288_I290C_FW (CGC CTT GAA AGA CTG TAT CAC GGT GGG) and TM288_I290C_RV (CCC ACC GTG ATA CAG TCT TTC AAG GCG).

Purified mutant proteins were all analyzed by SEC and did not differ in terms of elution profile and yield from the wild-type transporter.

**Crystallization.** For crystallization of TM287/288(EtoA) or TM287/288(2xDtoA/EtoA) in complex with Sb_TM#35, freshly purified transporter in 0.3% (w/v) β-DM was concentrated to about 12 mg/ml using an Amicon Ultra-4 concentrator unit (50 kDa MWCO) and purified Sb_TM#35 (stored at −80 °C) was added in a 1.1-fold molar excess (final complex concentration of 10 mg/ml). Complexes were pre-incubated with 2.5 mM adenosine 5′-(3-thiotriphosphate) (ATPγS, Sigma, A1388) or 5 mM ATP, 3 mM $MgCl_2$ for 5–6 days at 20 °C (without this incubation step, the crystals did not diffract to high resolution), before crystals were grown by the vapor diffusion method in sitting drops (1:1 protein to reservoir ratio) at 20 °C. Crystals were picked from wells containing either 0.1 M Na-acetate pH 4.6, 0.035 M NaCl and 11.5% (w/v) PEG6000 (ATPγS-bound structure), or 0.1 M Na-acetate pH 4.6, 0.025 M NaCl and 12% (w/v) PEG6000 (ATP-bound structure). Crystals appeared within 1–3 days and were fished immediately. Crystals were cryo-protected in 0.1 M Na-acetate pH 4.6, 0.04 M NaCl and 15% (w/v) PEG6000 additionally containing 20 mM Tris-HCl pH 7.5, 150 mM NaCl, 0.3% (w/v) β-DM, 3 mM $MgCl_2$, 1.25 mM ATPγS, or 5 mM ATP and 25% (v/v) ethylenglycol and flash-frozen in liquid nitrogen.

For crystallization of TM287/288(2xDtoA/EtoA) in complex with Nb_TM#1, purified Nb_TM#1 (stored at −80 °C) was added to the transporter purified in 0.3% (w/v) β-DM in a 1.2-fold molar excess prior to size-exclusion chromatography. After short incubation on ice, the complex was separated from excess nanobodies by size-exclusion chromatography using a Superdex 200 Increase 10/300 GL (GE Healthcare) column equilibrated in 20 mM Tris-HCl pH 7.5, 150 mM NaCl, and 0.3% (w/v) β-DM. The transporter/nanobody complexes were concentrated to 10 mg/ml using an Amicon Ultra-4 concentrator unit (50 kDa MWCO) and incubated with 5 mM ATPγS and 3 mM $MgCl_2$ for 15 min on ice. Crystals were grown by the vapor diffusion method in sitting drops (1:1 protein to reservoir ratio) at 20 °C in 0.05 M Glycine pH 9.5, 0.225 M NaCl and 21% (v/v) PEG550MME. Crystals appeared within 2–3 days and were grown for another 3 weeks. Crystals were cryo-protected in reservoir solution containing 10% (v/v) PEG400 and flash-frozen in liquid nitrogen.

For crystallization of TM287/288(2xDtoA/EtoA) in complex with Nb_TM#2, purified Nb_TM#2 (stored at −80 °C) was added to the transporter purified in 0.03% (w/v) β-DDM in a 1.2-fold molar excess prior to size-exclusion chromatography. After short incubation on ice, the complex was separated from excess nanobodies by size-exclusion chromatography using a Superdex 200 Increase 10/300 GL (GE Healthcare) column equilibrated in 20 mM Tris-HCl pH 7.5, 150 mM NaCl, and 0.03% (w/v) β-DDM. The transporter/nanobody complexes were concentrated to 10 mg/ml using an Amicon Ultra-4 concentrator unit (50 kDa MWCO) and incubated with 2.5 mM ATPγS and 3 mM $MgCl_2$ for 15 min on ice. Crystals were grown by the vapor diffusion method in sitting drops (1:1 protein to reservoir ratio) at 20 °C in 0.1 M Tris-HCl pH 8.5, 0.1 M NaCl and 30% (v/v) PEG400. Crystals appeared within 2–3 days and were grown for another 2–3 weeks. Crystals were flash-frozen in liquid nitrogen without further cryo-protection.

**Data collection and structure determination.** Diffraction data were collected with a wavelength of 1.0 Å at 100 K at the beamlines X06DA and X06SA at the Swiss Light Source (SLS, Villigen, Switzerland). Diffraction data were processed with the program XDS[39] and truncated using the Diffraction Anisotropy Server with default settings[40] due to strong or severe anisotropy, what lead to improved electron density maps (Supplementary Table 1, Supplementary Fig. 13).

The TM287/288(EtoA) – Sb_TM#35 – ATPγS-Mg complex structure was solved by molecular replacement in Phaser[41] using a modified homology model based on Sav1866 (PDB: 2HYD). The crystals belong to the space group P2₁ containing two TM287/288 heterodimers and two sybodies per asymmetric unit. After a few cycles of model building in Coot[42] and refinement in Buster (www.globalphasing.com), a poly-alanine model of a nanobody (PDB: 1ZVH) was manually placed into additional electron density. Multiple iterations of model building in Coot and TLS refinement in Buster resulted in a final model with good geometry (Ramachandran favored/outliers: 96.54%/0.08%) (Supplementary Table 1). Chains A (TM287), B (TM288) and E (Sb_TM#35) were used for structural analysis and figures.

In order to determine the TM287/288(2xDtoA/EtoA) – Sb_TM#35 – ATP-Mg complex structure, the final TM287/288(EtoA) – Sb_TM#35 – ATPγS-Mg complex structure was used for refinement against the TM287/288(2xDtoA/EtoA) – Sb_TM#35 – ATP-Mg data. The 2xDtoA mutations were introduced and the ATPγS replaced by ATP in Coot. TLS refinement in Buster resulted in a final model with good geometry (Ramachandran favored/outliers: 96.58%/0.24%) (Supplementary Table 1). Chains A (TM287) and B (TM288) were used for structural comparison with the TM287/288(EtoA) – Sb_TM#35 – ATPγS-Mg complex structure.

The TM287/288(2xDtoA/EtoA) – Nb_TM#1 – ATPγS-Mg complex structure was solved by molecular replacement in Phaser using the TM287/288(EtoA) –

Sb_TM#35 structure without the sybody. The crystals belong to the space group P2₁ containing two TM287/288 heterodimers and two nanobodies per asymmetric unit. After some cycles of model building in Coot and refinement in Buster, Phaser was used to place the missing nanobody using a homology model based on PDB entry 5OCL. Multiple iterations of model building in Coot and TLS refinement in Buster resulted in a final model with good geometry (Ramachandran favored/outliers: 94.68%/0.24%) (Supplementary Table 1). Chain A (TM287), B (TM288) and E (Nb_TM#1) were used for figures.

The TM287/288(2xDtoA/EtoA) – Nb_TM#2 – ATPγS-Mg complex structure was solved by molecular replacement in Phaser using the TM287/288(EtoA) – Sb_TM#35 structure without the sybody. The crystals belong to the space group P1 containing two TM287/288 heterodimers and two nanobodies per asymmetric unit. After several cycles of model building in Coot and refinement in Buster, Phaser was used to place the missing nanobody using a poly-alanine homology model based on PDB entry 4LAJ. Multiple iterations of model building in Coot and TLS refinement in Buster resulted in a final model at 4.2 Å resolution (Ramachandran favored/outliers: 93.34%/0.83%) (Supplementary Table 1). Chain A (TM287), B (TM288) and E (Nb_TM#2) were used for figures. Molecular graphics and analyses were performed in Pymol or with UCSF Chimera[43].

**ATPase assays.** ATPase activities were measured by detecting liberated phosphate using molybdate/malachite green method. To detect phosphate, reaction solution (90 μl) was mixed with filtrated malachite green detection solution (160 μl) consisting of 10.5 mg/mL ammonium molybdate, 0.5 M $H_2SO_4$, 0.34 mg/ml malachite green, and 0.1% Triton X-100, and absorption was measured at 650 nm[8]. ATPase activity measurements with detergent-purified protein were carried out in ATPase buffer consisting of 20 mM Tris-HCl pH 7.5, 150 mM NaCl, 10 mM $MgSO_4$ containing 0.03% (w/v) β-DDM. Activity measurements of TM287/288 reconstituted in nanodiscs were performed in the same buffer lacking detergent. Measurements of EfrEF reconstituted in proteoliposomes were carried out in 50 mM HEPES pH 7.0 and 10 mM $MgSO_4$.

Relative ATPase activities of TM287/288(D41A^TM287), TM287/288 (D65A^TM288), and TM287/288(2xDtoA) compared to wild-type TM287/288 were measured at 25 °C for 15 min in the presence of 500 μM ATP. 32 nM wild-type TM287/288, 64 nM single DtoA TM287/288 variants and 128 nM TM287/288 (2xDtoA) were used and the respective concentration of TM287/288(EtoQ) for background subtraction.

The relative ATPase activity stimulations of EfrEF variants in proteoliposomes by ethidium were determined at 30 °C for 15 min in the presence of 1 mM ATP. The amount of reconstituted EfrEF variants was determined by quantitative SDS-PAGE. 4 nM wild-type EfrEF, 23 nM single DtoA EfrEF variants, and 15 nM EfrEF (2xDtoA) were used and buffer controls were taken for background subtraction.

Inhibition of ATPase activities of wild-type TM287/288 and TM287/288 (2xDtoA) in detergent by binders was determined in the presence of 500 μM ATP at 25 °C for 30 min or 60 min, respectively. 8 nM wild-type TM287/288, as well as TM287/288(2xDtoA) were used, which is more than 2-fold less than the lowest binder concentration (20 nM). For background subtraction equal amounts of TM287/288(EtoQ) were used. To obtain IC₅₀ values, the inhibition data were fitted with a hyperbolic decay curve with the following function (SigmaPlot):

$$f = y_0 + (a \cdot IC_{50})/(IC_{50} + x) \tag{1}$$

in which $f$ corresponds to the ATPase activity at the respective binder concentration divided by the ATPase activity in the absence of inhibitor normalized to 100%, $y_0$ corresponds to the residual activity at infinite binder concentration, $a$ corresponds to the maximal degree of inhibition ($a + y_0 = 100\%$) and x corresponds to the binder concentration.

Stimulated ATPase activities of wild-type TM287/288 and TM287/288(2xDtoA) in nanodiscs were determined at 37 °C for 30 min in the presence of 500 μM ATP and varying Hoechst 33342 concentrations. 40 nM wild-type TM287/288, as well as TM287/288(2xDtoA) were used to determine relative stimulations compared to basal ATPase activities using buffer for background subtraction.

Hoechst 33342 stimulated ATPase activity inhibition in nanodiscs was determined at 37 °C for 30 min in the presence of 500 μM ATP and 50 μM Hoechst 33342. 8 nM wild-type TM287/288 and 40 nM TM287/288(2xDtoA) in nanodiscs were used to determine ATPase activities in the presence or absence of 10 μM binders using empty nanodiscs for background subtraction.

**BMOE cross-linking.** To raise alpaca nanobodies specifically recognizing the OF state of TM287/288, the transporter was cross-linked at the tetrahelix bundle, which forms when the transporter adopts the OF state[13]. Two cysteines were introduced in the cys-less TM287/288 variant at positions L200C^TM287 and S224C^TM288 by site-directed mutagenesis. On top, the 2xDtoA mutations were introduced in TM287/288_cl_L200C^TM287/S224C^TM288. Since the two cysteines are too far apart to form a disulfide bond, the maleimide cross-linker BMOE (bismaleimidoethane, Thermo Scientific™, #22323) with a length of 8 Å was used. TM287/288_cl_L200C^TM287/S224C^TM288 with or without the 2xDtoA mutations was expressed as described above. Membranes were solubilized and purified by Ni-NTA affinity chromatography in presence of 1 mM dithiothreitol (DTT) in 0.03% (w/v) β-DDM. Buffer was exchanged and DTT removed by size-exclusion chromatography using a Superdex 200 Increase 10/300 GL (GE Healthcare) column

equilibrated in PBS pH 7.4 and 0.03% (w/v) β-DDM at 4 °C and concentrated to 50 μM with an Amicon Ultra-4 concentrator unit with a MWCO of 50 kDa. 10 mM ATP, 3 mM $MgCl_2$, and a 5-fold molar excess of BMOE over transporter were added and the cross-linking mixture incubated for 3 h at 30 °C. The mixture was diluted 5-fold and incubated with 3 C protease (1:10 w/w) overnight at 4 °C. The cross-linked and Tag-free transporter was reloaded on a Ni-NTA gravity flow column equilibrated with 20 mM Tris-HCl pH 7.5, 150 mM NaCl, 40 mM imidazole and 0.03% (w/v) β-DDM. The sample was concentrated with Amicon Ultra-4 concentrator units with a MWCO of 50 kDa, 10% (v/v) glycerol was added and aliquots were flash-frozen in liquid nitrogen and stored at −80 °C ready for size-exclusion chromatography. Cross-linked TM287/288_cl_L200C^TM287/S224C^TM288 with or without the 2xDtoA mutations was polished by size-exclusion chromatography using a Superdex 200 Increase 10/300 GL (GE Healthcare) column equilibrated in 20 mM Tris-HCl pH 7.5, 150 mM NaCl and 0.03% (w/v) β-DDM at 4 °C and concentrated to 1 mg/ml with an Amicon Ultra-4 concentrator unit with a MWCO of 50 kDa and immediately used for alpaca immunizations.

**Nanobody and sybody selections.** For the selection of OF state-specific nanobodies, an alpaca was immunized with subcutaneous injections four times in two week intervals, each time with 200 μg purified cross-linked TM287/288(2xDtoA)_cl_L200C^TM287/S224C^TM288 in 20 mM Tris-HCl pH 7.5, 150 mM NaCl, and 0.03% (w/v) β-DDM. Immunizations of alpacas were approved by the Cantonal Veterinary Office in Zurich, Switzerland (animal experiment licence nr. 188/2011). Blood was collected two weeks after the last injection for the preparation of the lymphocyte RNA, which was then used to generate cDNA by RT-PCR to amplify the VHH/nanobody repertoire. Phage libraries were generated and two rounds of phage display were performed against TM287/288(2xDtoA/EtoA) solubilized in β-DDM in the presence of 2 mM ATP-Mg. After the final phage display selection round, 91.9-fold enrichment was determined by qPCR using AcrB as background. The enriched nanobody library was subcloned into pSb_init by FX cloning and 94 single clones were analyzed by ELISA in the presence of 1 mM ATP-Mg. The 27 positive ELISA hits were Sanger sequenced and grouped in three families according to their CDR3 length and sequence (among them was Nb_TM#1). In another selection cross-linked TM287/288_cl_L200C^TM287/S224C^TM288 was used for alpaca immunizations resulting in 51 positive ELISA hits, of which 24 were Sanger sequenced and grouped into four binder families (among them was Nb_TM#2).

Sybodies were selected against TM287/288(E517A) in β-DDM in presence of 1 mM ATP-Mg with our in vitro selection platform[14]. After the second round of phage display, single clones were analyzed for binding against TM287/288(E517A) in presence of ATP-Mg by ELISA. Sequencing of 48 ELISA positives resulted in 40 unique sybody sequences[14].

**Spin labeling for DEER.** TM287/288 cysteine variants were expressed as described above and purified by Ni-NTA affinity chromatography in presence of 2 mM DTT. For spin labeling, DTT was removed on a PD-10 column (GE Healthcare, 17-0851-1) equilibrated with 20 mM Tris-HCl pH 7.5, 150 mM NaCl, and 0.03% β-DDM and MTSL [(1-oxyl-2,2,5,5-tetramethyl-Δ3-pyrroline-3-methyl)methanethiosulfonate, Toronto Research] was added in a 10-fold molar excess and incubated at 4 °C overnight. Free spin-label was removed by size-exclusion chromatography on a Superdex 200 Increase 10/300 GL (GE Healthcare) column equilibrated in 20 mM Tris-HCl pH 7.5, 150 mM NaCl, and 0.03% (w/v) β-DDM at 4 °C. The samples were concentrated to 30–50 μM with Amicon Ultra-4 concentrator units with a MWCO of 50 kDa, flash-frozen in liquid nitrogen and stored at −80 °C ready for DEER experiments. The pairs 131^TM288/248^TM288 and 460^TM287/363^TM288 were already used in previous studies[8,9], whereas the extracellular spin-label pairs 54^TM287/271^TM287 and 54^TM287/290^TM288 were constructed as part of this study and their ATPase activities were determined (Supplementary Table 3).

For site-specific spin labeling of Sb_TM#35, a single cysteine was introduced in the framework of the sybody at position 71 by site-directed mutagenesis. Sb_TM#35_S71C was expressed from pBXNPHM3 as described above. Cells were harvested and resuspended in PBS pH 7.4 and 2 mM DTT supplemented with DNase (Sigma) and disrupted with an M-110P Microfluidizer® (Microfluidics™). The supernatant, supplemented with 20 mM imidazole, was loaded on a Ni-NTA gravity flow column, washed with 20 column volumes PBS pH 7.4, 50 mM imidazole and 2 mM DTT and eluted with 4 column volumes PBS pH 7.4, 300 mM imidazole and 2 mM DTT.

In a next step Sb_TM#35_S71C fused to His-tagged MBP was incubated with 3 C protease (1:10 w/w), while dialyzing against PBS pH 7.4 and 2 mM DTT overnight at room temperature. Cleaved sybody was reloaded on a Ni-NTA gravity flow column and eluted with three column volumes PBS pH 7.4, 40 mM imidazole and 2 mM DTT, followed by size-exclusion chromatography using a Sepax-SRT10C SEC-300 (Sepax Technologies) column equilibrated with PBS pH 7.4 and 2 mM DTT. Peak fractions were collected and DTT removed using a PD-10 column (GE Healthcare, 17-0851-1) equilibrated with degassed PBS pH 7.0 at 4 °C. To avoid DTT take-over, the sybodies were eluted with 3.2 ml degassed PBS pH 7.0, instead of the 3.5 ml suggested by the manufacturer. The elution was concentrated to 2.5 ml using an Amicon concentrator unit with a 3 kDa MWCO. 5-fold molar excess MTSL was added to the sample and incubated for 1 h on ice, a condition which was previously reported to prevent labeling of the buried cysteines that form the universally conserved disulfide bond of nanobodies[44]. Free label was

removed and buffer exchanged using a PD-10 column (GE Healthcare, 17-0851-1) equilibrated with 20 mM Tris-HCl pH 7.5 and 150 mM NaCl. In order to avoid MTSL take-over, the sybodies were eluted with 3.2 ml 20 mM Tris-HCl pH 7.5 and 150 mM NaCl. Spin-labeled sybodies were concentrated to the desired concentration with an Amicon Ultra-4 concentrator unit with a 3 kDa MWCO and flash-frozen in liquid nitrogen and stored at −80 °C. Site-specific labeling was confirmed and quantified by mass spectrometry.

**DEER measurements.** The labeling efficiency of the double cysteine mutants of the transporters solubilized in detergent was measured by comparing the second integral of the spectra detected at 25 °C using an X-band Miniscope 400 EPR spectrometer (Magnettech by Freiberg Instruments) with that of a standard TEMPOL solution in water. The calculated spin labeling efficiencies of the twelve mutants ranged between 80% and 90%. For DEER measurements, 10% (v/v) D$_8$-glycerol was added as cryoprotectant. The range of final transporter concentrations was 15 to 25 μM. The sample (40 μl) was loaded in quartz tubes with 3 mm outer diameter. The ATP-EDTA sample contained 2.5 mM ATP and 2.5 mM ethylenediaminetetraacetate (EDTA) to completely inhibit ATP hydrolysis; samples were incubated at 25 °C for 10 min and snap-frozen in liquid nitrogen. For vanadate trapping, samples were incubated with 5 mM sodium orthovanadate, 2.5 mM ATP and 2.5 mM MgCl$_2$ for 3 min at 50 °C and snap-frozen in liquid nitrogen. The unlabeled Sb_TM#35 was added to the TM287/288 in ~1.3:1 stoichiometric ratio. To measure sybody-transporter distances, Sb_TM#35 spin-labeled at position 71 was added in a 0.5:1 stoichiometric ratio to the singly-labeled TM287/288 mutants (54$^{TM287}$ and 271$^{TM287}$) in the presence of ATP-EDTA.

Double electron–electron resonance (DEER) measurements were performed at 50 K on a Bruker ELEXSYS E580Q-AWG (arbitrary waveform generator) pulse Q-band spectrometer equipped with a 150 W TWT amplifier. A 4-pulse DEER sequence with Gaussian, non-selective observer and pump pulses of 32 or 34 ns length (corresponding to 14 or 16 ns FWHM) with 100 MHz frequency separation was used. Due to the coherent nature of the AWG generated pulses, a four-step phase cycling (0−π/2−π−3/2π) of the pump pulse was performed together with 0−π phase cycling of the observer pulses to remove unwanted effects of running echoes from the DEER trace. The evaluation of the DEER data was performed using DeerAnalysis2015[45]. The background of the primary DEER traces was corrected using stretched exponential functions with homogeneous dimensions of 1.8 to 3 for different samples. A model-free Tikhonov regularization was used to extract distance distributions from the background corrected form factors. The data of the apo and ATP-EDTA states shown for the pairs 131$^{TM288}$/248$^{TM288}$ and 460$^{TM287}$/363$^{TM288}$ in the absence of sybody are reproducible with respect to those previously published[9]. Interspin distance simulations were performed with the software MMM2015 using the MTSL ambient temperature library[46,47].

**Transport assay.** *L. lactis* NZ9000 Δ*lmrA*Δ*lmrCD* cells harboring the plasmids of wild-type or mutant EfrEF were grown in M17 medium supplemented with 0.5% glucose and 5 μg/ml chloramphenicol to an OD$_{600}$ of 0.4–0.6 at 30 °C and the expression was induced by adding a nisin-containing culture supernatant of *L. lactis* NZ9700 for 2 h (1:1000 [v/v]). Cells were washed and resuspended using fluorescence buffer (50 mM KP$_i$ pH 7.0 and 5 mM MgSO$_4$). Cells were diluted to an OD$_{600}$ of 0.5 and energized by adding 0.5% glucose. The accumulation of 5 μM ethidium was monitored at 30 °C using a Fluorescence Spectrometer LS-55 (Perkin Elmer, Schwerzenbach, Switzerland). Excitation and emission wavelengths (and slit widths) were set at 520 nm (10 nm) and 595 nm (15 nm)[16,17].

**Reconstitution into proteoliposomes.** *E. coli* polar lipids extracted from *E. coli* total lipids (Avanti lipids 100500 P) and L-α-Phosphatidylcholine (from egg yolk, Type XVI-E, ≥ 99% (TLC) P3556 Sigma) were dissolved in chloroform. Lipids were mixed in a 3:1 (w/w) ratio, chloroform was evaporated and dried lipids were dissolved in reconstitution buffer (50 mM K-HEPES pH 7.0). Lipids were sonicated to generate small unilamellar vesicles (SUVs). SUVs were flash-frozen in liquid nitrogen and thawed four times to fuse the SUVs to large multilamellar vesicles (LMVs). Large unilamellar vesicles (LUVs) were finally formed by extruding LMVs through a 400 nm polycarbonate filter. LUVs were diluted to a working concentration of 4 mg/ml and destabilized using 5.25 mM Triton X-100. Detergent-purified EfrEF variants were added to the destabilized liposomes at a protein:lipid ratio of 1:100. Detergent molecules were removed by four rounds of adding and removing 200 mg Bio-Beads (SM-2 polystyrene beads, Bio-Rad). Proteoliposomes were harvested by centrifugation (40'000 rpm, 70 Ti rotor, Beckman) and resuspended in 50 mM K-HEPES pH 7.0[16].

**Surface plasmon resonance.** Binding affinities were determined by surface plasmon resonance at 25 °C using a ProteOn™ XPR36 Protein Interaction Array System (Biorad). Biotinylated TM287/288 variants were immobilized on ProteOn™ NLC Sensor Chips at a density of 2000 RU. Nanobodies and sybodies expressed in pSb_init were gel-filtrated in 20 mM Tris-HCl pH 7.5 and 150 mM NaCl, and the SPR measurements were carried out in the same buffer containing 0.015% (w/v) β-DDM and either 1 mM MgCl$_2$ or 1 mM MgCl$_2$ and 0.5 mM ATP to measure binding affinities in the absence or presence of ATP, respectively. Every measurement was done once and the data fitted with a 1:1 interaction model using the

BioRad Proteon Analysis Software. In order to determine the half-lives of the OF ATP-bound state of the different TM287/288 variants, all five biotinylated variants were immobilized on a ProteOn™ NLC Sensor Chips at a density of 3000 RU. The experiment was conducted in 20 mM Tris-HCl pH 7.5, 150 mM NaCl, 0.015% (w/v) β-DDM, and 1 mM MgCl$_2$ or 2.5 mM EDTA at 25 °C at a flow-rate of 30 μl/min. In order to charge the transporter variants with ATP-Mg or ATP-EDTA, buffer containing 1 mM ATP together with either 1 mM MgCl$_2$ or 2.5 mM EDTA was injected at the beginning of the experiment.

**Nanodisc preparation.** Membrane scaffold protein MSP1E3D1 was subcloned from pINITIAL (provided by Prof. Raimund Dutzler) into pBXNH3 (addgene: Plasmid #47067) by FX cloning[36]. Freshly transformed MC1061 *E. coli* cells were grown in Terrific Broth (TB) medium supplemented 100 μg/ml ampicillin to an OD$_{600}$ of 1.0–1.5 at 37 °C and expression was induced by the addition of 0.0017% (w/v) L-arabinose overnight at 22 °C. Cells were harvested, resuspended in lysis buffer (20 mM Na-phosphate pH 7.4, 1% (v/v) Triton X-100 and 1 mM PMSF), and disrupted with a M-110P Microfluidizer® (Microfluidics™). Cell debris were pelleted at 8000 g for 30 min at 4 °C and the supernatant was loaded on a Ni-NTA gravity flow column equilibrated with lysis buffer, washed with 10 column volumes buffer 1 (40 mM Tris-HCl pH 8.0, 0.3 M NaCl and 1% (v/v) Triton X-100), 10 column volumes buffer 2 (buffer 1 + 50 mM sodium cholate), 10 column volumes buffer A (40 mM Tris-HCl pH 8.0 and 0.3 M NaCl), 10 column volumes buffer A containing 20 mM imidazole, and finally eluted with 4 column volumes buffer A containing 300 mM imidazole. 3 C protease was added (1:10 w/w) and the sample dialyzed overnight at 4 °C against 20 mM Tris-HCl pH 7.5, 150 mM NaCl, and 0.5 mM K-EDTA. The next day, 2 mM MgCl$_2$ were added to the sample, which was then reloaded on a Ni-NTA gravity flow column equilibrated with 20 mM Tris-HCl pH 7.5, 150 mM NaCl, and 20 mM imidazole and eluted with two column volumes using the same buffer. Buffer was exchanged by dialyzing three times against Tris-HCl pH 7.5 and 0.5 mM K-EDTA for 1.5 h at room temperature. The purified membrane scaffold protein was concentrated to 60 mg/ml using an Amicon Ultra-4 concentrator unit with a 10 kDa MWCO, flash-frozen in liquid nitrogen, and stored at −80 °C.

*E. coli* polar lipids (*E. coli* polar lipid extract, Avanti, 100600C) were mixed 3:1 (w/w) with L-α-phosphatidylcholine from egg yolk (Sigma, P3556) and chloroform was evaporated. Dried lipids were dissolved in 20 mM HEPES pH 8.0, 0.5 mM K-EDTA, 100 mM NaCl, and 100 mM cholate to a final concentration of 38 mg/ml (50 mM), and filtered using a 0.22 μM filter. Lipids ready for nanodisc reconstitutions were stored at −80 °C.

Purified, biotinylated TM287/288 variants solubilized in β-DDM were reconstituted into nanodiscs using a 240:8:1 molar ratio of lipids:MSP1E3D1:TM287/288. In order to spare membrane protein, the ideal lipid:MSP1E3D1 ratio was determined beforehand by reconstituting empty nanodiscs. Different ratios were tested, ranging from 25:1 to 40:1 (lipid:MSP1E3D1). Empty nanodiscs were loaded on a Superdex 200 Increase 10/300 GL (GE Healthcare) column to separate empty nanodiscs from monomeric MSP1E3D1 and aggregates. From the elution profile the optimal lipid:MSP1E3D1 ratio of 30:1 was determined. The final cholate concentration in the reconstitution mixture was adjusted to 30 mM. The mixture was incubated at 25 °C for 20 min while rocking at 650 rpm. Then, 200 mg bio-beads were added to 200 μl reconstitution mixture, which was incubated overnight at 4 °C while rocking at 1000 rpm. Bio-beads were removed using 0.1 μm PVDF spin-filters. Full nanodiscs were separated from empty nanodiscs by gel filtration using a Superdex 200 Increase 10/300 GL (GE Healthcare) equilibrated with 20 mM Tris-HCl pH 7.5 and 150 mM NaCl. The SEC profile as well as a SDS-PAGE analysis of nanodisc-reconstituted TM287/288 is shown in Supplementary Fig. 2c. TM287/288 variants reconstituted in nanodiscs were either immediately used or flash-frozen in liquid nitrogen and stored at −80 °C.

**MD simulations.** The simulation setup and parameters are equivalent to our previous work[11]. In brief, all-atom MD simulations of TM287/288 embedded in an explicitly solvated POPC bilayer were carried out with GROMACS[48]. It should be noted that the source organism of TM287/288, the hyperthermophilic bacterium *Thermotoga maritima*, has a unique lipid composition[49], which is difficult to implement for MD simulations. For the simulations initiated from the IF structure (PDB: 4Q4A), ATP-Mg was docked into the consensus site and D41$^{TM287}$ and D65$^{TM288}$ were replaced by alanines. We conducted 20 individual simulations of 500 ns each (i.e., 10 μs in total) at 375 K. In addition, the OF state was simulated starting from the crystal structure (PDB: 6QUZ) shown in Fig. 1a after removing the sybody and replacing ATPγS by ATP. Furthermore, the same set of simulations was carried out for the ATP-bound OF structure in which the 2xDtoA mutations were introduced in silico (another 8 μs in total). In addition, the OF wild-type X-ray structure was simulated in a POPE (instead of POPC) bilayer at 375 K (ten simulations of 500 ns each, i.e., 5 μs in total).

**Reporting summary.** Further information on research design is available in the Nature Research Reporting Summary linked to this article.

## Data availability

Data supporting the findings of this manuscript are available from the corresponding authors upon reasonable request. A reporting summary for this Article is available as a Supplementary Information file. The source data underlying Figs. 2b, 4c, d, g and Supplementary Figs. 2a, b are provided as a Source Data file. The coordinates of the TM287/288 structures have been deposited under accession numbers 6QUZ (Sb_TM#35, ATPγS-bound), 6QV0 (Sb_TM#35, ATP-bound), 6QV1 (Nb_TM#1) and 6QV2 (Nb_TM#2). Sybody Sb_TM#35 and nanobodies Nb_TM#1 and Nb_TM#2 will be distributed for academic research upon reasonable request.

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

## Acknowledgements

We thank all members of the Seeger lab for stimulating discussions. We acknowledge Beat Blattmann and Céline Stutz-Ducommun of the Protein Crystallization Center UZH for performing the crystallization screening, and the staff of the SLS beamlines X06SA and X06DA for their support during data collection. E.B. would like to thank G. Jeschke (ETH Zurich) for providing the Q-band resonator. The Steinbuch Centre for Computing (SCC) in Karlsruhe/Germany provided computational resources. The Institute of Medical Microbiology and the University of Zurich are acknowledged for financial support. M.K. thanks Natural Sciences and Engineering Research Council of Canada (NSERC) and the Canada Research Chairs Program for financial support. This work was funded by a SNF Professorship of the Swiss National Science Foundation (PP00P3_144823, to

M.A.S.) and by the Deutsche Forschungsgemeinschaft (DFG, German Research Foundation) under Germany's Excellence Strategy—EXC-2033—project number 390677874, Emmy Noether grant to L.V.S. (SCHA 1574/3-1), and research grant to E.B. (BO 3000/1–2 and INST 130/972-1 FUGG).

## Author contributions

C.A.J.H., E.B., and M.A.S. conceived the study. C.A.J.H., I.Z., P.E., and S.S. selected nanobodies and sybodies against TM287/288. C.A.J.H. purified and crystallized the protein complexes and solved their structure. L.M.H. conducted the functional and biochemical experiments with EfrEF. C.A.J.H. and L.M.H. conducted all functional experiments with TM287/288. C.A.J.H. cloned, purified, and labeled all samples for DEER analyses. M.H.T. conducted most of the DEER analyses and simulations and discussed the results with E.B.. S.K. conducted the sybody-transporter DEER experiments and discussed them with E.B.. H.G. carried out MD simulations and analyzed and interpreted them together with M.K. and L.V.S.. C.A.J.H., M.H.T., L.M.H., H.G., and M.A.S. created figures. C.A.J.H., L.V.S., E.B., and M.A.S. wrote the manuscript and all authors edited the manuscript.

## Additional information

**Competing interests:** The authors declare no competing interests.

