## [Peer Review File · Nature Communications]

Reviewers' comments:

Reviewer #1 (Remarks to the Author):

This manuscript describes detailed structural and function work on a heterodimeric ABC exporter shifted to the outward-phasing state by a sybody. PELDOR/DEER data are of high quality and significance and clearly prove that this conformational shift happens. Several important conclusions about the transport and inhibition mechanism can be drawn from this data. Mutants show that extracellular gate closure is required for facilitating NBD dimer separation and thus transport cycling. All findings and the thereof drawn conclusions are convincing. Combining structural, spectroscopic and simulation results gives very interesting new insights into the function of this important class of heterodimeric membrane - therefore I recommend publication of this article in Nature Communication

Reviewer #2 (Remarks to the Author):

Hutter et al provide a structural and biophysical analysis of the archaeal, heterodimeric ABC transporter TM287/288. The authors present crystal structures of outward-open TM287/288, one with a synthetic single domain antibody (sybody) bound on the extracellular side and two with nanobodies bound to the nucleotide binding domains. The structures are in complex with a non-hydrolyzable nucleotide analogue, ATPS. The authors say that the sybody was essential for achieving high resolution of the outward-open structure (3.2 Å). They propose a mechanism in which the sybody allosterically inhibits the transporter and attempt to use it as a probe to show that closure of the extracellular gate is required for dissociation of the NBD dimer. They include some molecular dynamics analysis, mutagenesis data, and DEER and SPR measurements to attempt to support their mechanistic proposals.

We doubt the usefulness and impact of the study due to the following:

- The fundamental point / assumption of the study is that TM287/288, an archaeal 'model' system can provide valuable insight into eukaryotic ABC exporters including MRP1 or CFTR (lines 331-333). In our view, this concept is invalid. First, although the authors assume that TM287/288 is a drug transporter, there is no evidence supporting this (see also below). Its role in transporting drugs in the native organism (*T. maritima*) has not been demonstrated. It is therefore very unlikely that insight gained from this prokaryotic ABC transporter of unknown physiological function can be applied to any of the eukaryotic systems. Second, the authors themselves demonstrate that there is major heterogeneity in the degree of extracellular gate opening amongst ABC exporters (lines 281-291 and Fig. S6), exemplifying significant mechanistic differences. If the authors want to test their hypotheses and draw conclusions concerning the mechanism of MRP1 or CFTR, they should study these proteins directly rather than relying on an unjustified model system.
- The authors report an outward-facing state of TM287/288, but given that this is not the first outward-open structure, it is unclear how this is relevant for the broader scientific audience. Is it to understand substrate transport? If so which substrates are transported? In the abstract (lines 16 and 17) the authors say that for ABC exporters, 'extracellular gate opening and closure are key steps of

the transport cycle', yet this concept is completely unproven.

We conclude that the study does not advance our mechanistic understanding of this ABC transporter (because we don't know what it is supposed to transport) nor of ABC transporters in general because there is no evidence supporting the idea that conclusions can be drawn for eukaryotic systems. It is a pity the authors did not dedicate their efforts to an ABC transporter of known function, where biophysical insight could be used to explain functional properties.

In addition to the major criticism above and while the study may have technical merits and may prompt similar studies of better-characterized ABC systems, there are unfortunately numerous issues with the interpretation (or over-interpretation) of data, as well as too much speculation. We include a point-by-point list of these concerns below:

- Line 19: In the abstract, the authors say that the sybody recognizes TM287/288 exclusively in the presence of ATP. On line 117 they say that the fully dimerized NBDs sandwich binds two ATP molecules. Actually, ATPS is bound - This is an important difference seeing as in a previous structure of TM287/288, and supported by subsequent DEER studies, the transporter remained in an inward-facing state when one AMP-PNP molecule was bound to the degenerate catalytic site (see references below). How can the authors be sure that the structure with ATPS bound will be the same as that in complex with ATP?

Reference: Crystal structure of a heterodimeric ABC transporter in its inward-facing conformation; Hohl et al, NSMB. 2012 March 25; 19(4):395-402. doi: 10.1038/nsmb.2267.

Reference: Exploring conformational equilibria of a heterodimeric ABC transporter; Timachi et al, eLife. 2017 Jan 4;6: pii: e20236. doi: 10.7554/eLife.20236.

- In the abstract and throughout the manuscript (lines 146, 178, 214, 286) the authors state that the sybody shifts the transporters conformational equilibrium to the outward-facing state. In Figs. 2c and 4f, they show that in the presence of ATP-EDTA, Sb_TM#35 promotes a more outward-facing conformation. However, in the absence of ATP the sybody does not bind. Is it simply a case that the sybody does not allow full closure of the extracellular side after it binds? If so then why would this inhibit ATP hydrolysis?

- Several state-specific sybodies were generated for TM287/288 before, including Sb_TM#35 (see reference below). In this previous study, it was demonstrated that another sybody, Sb_TM#26, binds to both wild type and TM287/288_E517A in the presence of ATP-Mg with a higher affinity than Sb_TM#35. Why did the authors choose Sb_TM#35 for structural studies?

Reference: Synthetic single domain antibodies for the conformational trapping of membrane proteins; Zimmermann et al, eLife. 2018 May 24; 7: pii: e34317. doi: 10.7554/eLife.34317.

- Lines 70-71 and 275: The authors state that single domain antibodies were instrumental/ essential for solving the structure in an outward-facing state. We assume the authors attempted to trap an outward-facing state in the absence of a sybody or nanobody. On line 194, and shown by molecular dynamics simulations in Fig. S11, the authors state that the outward-facing conformation is very stable in the absence of sybody. We appreciate that Fig. S1 shows that the sybody forms crystal contacts, but there are a number of ABC exporter structures in outward-facing conformations,

determined at higher resolutions by either X-ray crystallography or cryo-EM, in the absence of conformational binders.

- Line 79: The authors acknowledge their previous work showing that a Walker B variant (TM287/288 E517Q) was almost completely trapped in an outward-facing state in the presence of ATP-Mg or ATPS-Mg (eLife reference below). The ATPase activity of this variant was reported to be extremely low (0.165 nmolPi min⁻¹ mg⁻¹, which was 0.19% of the wild type activity). Was the EtoQ variant not a valid structural target? Was it necessary to reduce the activity any further by introducing the EtoA mutation? We appreciate that Fig.5b (and explained in lines 240-257) shows the t_{1/2} of the outward-facing state is generally longer for the EtoA variant, but these experiments were not done with ATPS (used for the structure). Why did the authors use ATPS-Mg? What is the t_{1/2} of the variants in the presence of ATPS-Mg? These points are important to address seeing as TM287/288 responds differently to different nucleotide analogues (see references below).

Reference: Exploring conformational equilibria of a heterodimeric ABC transporter; Timachi et al, eLife. 2017 Jan 4;6. pii: e20236. doi: 10.7554/eLife.20236.

Reference: Crystal structure of a heterodimeric ABC transporter in its inward-facing conformation; Hohl et al, NSMB. 2012 March 25; 19(4):395-402. doi: 10.1038/nsmb.2267.

- The authors see very different effects in detergent and nanodisc environments. Fig. 2b shows that in detergent Sb_TM#35 inhibits ATPase activity by ~80%, at a concentration of 1 μM, and has an IC₅₀ of 66 nM. Fig. S2b shows that in nanodiscs Sb_TM#35 only inhibits by ~40% even though it was added at a much higher concentration (10 μM). Why was 10 μM Sb_TM#35 added in Fig. S2b? Is there still inhibition at the concentrations used in Fig. 2b? How do the raw ATPase activities (nmolPi min⁻¹ mg⁻¹) compare in detergent and nanodiscs? Have the binders been tested in proteoliposomes? These comparisons are important seeing as it was previously reported that TM287/288 has ~7-fold lower ATPase activity when reconstituted in E.coli lipid/PC liposomes compared to detergent (see reference below)

Reference: Crystal structure of a heterodimeric ABC transporter in its inward-facing conformation; Hohl et al, NSMB. 2012 March 25; 19(4):395-402. doi: 10.1038/nsmb.2267.

- The authors make a statement on line 100 that, 'hence, the sybody addresses an epitope that is accessible by biopharmaceuticals in the cellular context.' The authors need to explain what they mean by this statement, especially in light of the disparity between their detergent and nanodisc results and lack of assessment in proteoliposomes, as mentioned above.

- Lines 147, 176, 239, Figs. 2c, 4f, 5b: Why was ATP-EDTA used to create non-hydrolyzing conditions when ATPS was used for the structures?

- Line 150 and Fig. 2c: The authors state that sybody binding changes the distance between two spin labels positioned in the wing underneath the sybody. Do the authors think that this distance change has any mechanistic implications?

- Lines 141 and 165: The authors call the sybody a 'molecular clamp', but Fig. 2a does not show this clearly. A 90 rotation is also required to demonstrate how the sybody acts as a molecular clamp. The authors could include the sybody in Fig. S4 (we assume the sybody has been removed here but this

is not stated in the legend).

- A number of TM287/288 variants are presented in this study for both structural and functional purposes. It would be beneficial to show SEC profiles of these mutants compared to the wild type protein and to make a comment on their stability. We appreciate that the binding affinity of Sb_TM#35 and Nb_TM#1 is actually higher for the EtoA and 2xDtoA variants. However, the ATPase activities and percentage inhibition by Sb_TM#35 for the spin label variants is lower than for the wild type protein (Table S3).
- The authors use different variants for different structures. TM287/288 in complex with Sb_TM#35 is with an EtoA variant, while both nanobody complexes contain a 2xDtoA/ EtoA mutation. Can the authors explain why different variants were used? How can the authors be sure that it is the sybody, rather than the type of mutation, that explains the difference in resolution of each complex?
- Lines 196-211 and Fig. 4d,e: Why did the authors only study membrane reconstituted EfrEF and not TM287/288?
- Lines 258-262: Is it only data on TM287/288 for which these two major conclusions can be drawn? What about results from other ABC transporters?
- Lines 265-267: As mentioned in the points above, we feel the choice of target was a poor one. The authors should use their state-specific antibody approach to explore the mechanisms of ABC transporters of known function. Have the authors explored the function of TM287/288? What is the effect of a TM287/288 knockout/knockdown?
- Line 278: The authors say, 'this means that ATP hydrolysis of only one nucleotide is sufficient to initiate dissociation of the NBDs'. Do they have any proof of this?
- Lines 281-291 and Fig. S6: The authors compare structures of different ABC exporters and acknowledge major heterogeneity in the degree of extracellular gate opening. This exemplifies critical differences amongst ABC exporters and highlights how TM287/288 should not be used as a 'model' to understand other systems. Cross-talk between the extracellular gate and the ATPase cycle is not a general phenomenon of ABC transporters and may be specific to TM287/288.
- Lines 292-293: The authors say that using their nanobodies as conformational probes they were able to show that, 'extracellular gate closure is coupled to dissociation of the closed NBD dimer after substrate release and ATP hydrolysis'. We do not see how the nanobodies reveal this order of events (substrate release before extracellular gate closure). Surely this is an assumption as stated on line 326, 'it is reasonable to assume that the hydrolysis event happens after substrate release.'
- Lines 307-311: Interesting mechanistic speculation, but may only be applicable to TM287/288.
- Lines 324-326: Is TM287/288 really a drug transporter? Showing ATPase stimulation by Hoechst is completely inadequate (Fig. S2) as many other unrelated transporters transport or interact with this compound. Before TM287/288 can be called a "drug transporter", its role in transporting drugs in

the native organism (*T. maritima*) must be proven. Perhaps it is not a drug transporter at all but a translocase of cell wall components or O-antigens?

- Lines 330-333: There is no evidence that the results in this study provide a mechanistic framework to understand other ABC exporters. Either test the hypotheses on MRP1 and CFTR directly or do not mention these transporters at all.
- Lines 383-393: For crystallization TM287/288 was first mixed with Sb_TM#35 before ATPS and MgCl₂ were added. The authors write that ATPS-Mg had to be pre-incubated with the complex for 5-6 days at 20 C before crystals were grown in order to obtain high resolution. Seeing as Sb_TM#35 only recognizes the outward-facing transporter, would it not have made more sense to first incubate TM287/288 with ATPS-Mg and then add Sb_TM#35?
- Lines 394-404: Following on from the point above, the same is true for the complex with Nb_TM#1. Why was nanobody added before ATPS? In addition, why was the complex only incubated for 15 min prior to crystal growth when it is claimed above that a 5-6 day incubation was necessary for high resolution of the Sb_TM#35 complex? In the manuscript (lines 86 and 104) the authors say that the complex with Nb_TM#1 did not diffract well enough to build a reliable model (3.5 Å compared to 3.2 Å for Sb_TM#35). Is this simply due to the differences in sample preparation rather than any benefit of the sybody over the nanobody?
- Lines 383-415: Why were different concentrations of ATPS used for the different complexes (2.5 mM for Sb_TM#35 and Nb_TM#2, 5 mM for Nb_TM#1)? Why was a preparative SEC step used to remove excess nanobody but was not used to remove excess sybody?
- Line 499: Why was ATP added for the alpaca immunizations yet ATPS was used for the structures? Why was the ATP:MgCl₂ ratio 3.3:1?
- Lines 613-650: Why did the authors choose MSP1E3D1 (the authors write MSP1D1E3 but mean MSP1E3D1)? The authors state that, 'to spare membrane protein the ideal lipid:MSP ratio was determined beforehand by reconstituting empty nanodiscs,' (lines 638-639). Surely TM287/288 has to be included in the optimization procedure for the best lipid:MSP:TM287/288 ratio to be determined. SDS-PAGE and SEC profiles of the complexes should be provided to aid the explanations in this section.
- Fig. 1d: the figure legend is the wrong way around. SPR analyses in the absence of ATP is the upper panel and in the presence of ATP is the lower panel.
- Fig. 2b: can the authors explain why Nb_TM#1 only inhibits ATPase activity in detergent by ~40%? Seeing as it binds to the closed NBD dimer, would the authors not expect it to have a stronger inhibitory effect?

Reviewer #3 (Remarks to the Author):

The manuscript by Hutter et al reports on their study of bacterial ABC exporter TM287/288 using a novel synthetic single domain antibody that recognizes the transporter in the presence of ATP. They have been able to solve the crystal structure of the transporter in its OF state, which is significant since the OF structure of the transporter was not known before. They also use DEER, mutational studies, and molecular dynamics to support their main hypothesis that the "efficient extracellular gate closure is required to dissociate the NBD dimer after ATP hydrolysis".

The work is quite interesting in successful use of a synbody to capture the OF state of TM287/288, and in combining crystallography, DEER, biochemical, and computational data to build a comprehensive account of chemomechanical couplings in ABC exporters. The work certainly adds something to our knowledge of how ABC exporters work. Enough evidence has been provided for the main hypothesis of the paper; however, there are points that need to be addressed:

- 1) The generality of the finding. The authors should discuss their main finding within the context of ABC exporters in general. For instance, how the DEER data provided here compares to that reported for other ABC exporters such as MsbA, BmrCD, or P-gp.
- 2) The significance of the finding within the physiological context. The authors show that the synbody inhibits the ATPase activity by shifting the equilibrium towards the OF state. This is interesting but what does it mean in terms of the transport mechanism in the actual physiological context? This needs to be discussed.
- 3) The simulations were performed in a POPC bilayer. However, TM287/288 belongs to a gram-negative bacterium. I assume if one decides to simulate such a protein in a pure lipid bilayer, POPE would be the right choice. The authors should explain the reason for their choice and comment on how much they think this choice would have an effect on their simulation outcomes.
- 4) Minor point: MD is mentioned by I can't find it being defined, which is supposed to be Molecular Dynamics.

We thank the reviewers for their thoughtful and thorough comments and suggestions. We appealed to the initial rejection of this manuscript because we disagree with the main arguments of reviewer#2 regarding the usefulness of this study for the ABC transporter field. Rather, we feel that our analysis contains novel mechanistic aspects that will contribute to the mechanistic understanding of the poorly characterized re-setting step of the transport cycle, in which extracellular gate closure and its coupling to the NBD dimer appears to play a key role in TM287/288 and also EfrEF. In particular the closed NBD dimer of outward-facing ABC transporters is a conserved structural hallmark and its dissociation after ATP hydrolysis represents a mechanistic “problem” of general relevance. We have tamed/reworded our statements regarding eukaryotic ABC transporters to avoid misunderstandings. Apart from the fundamental critics of our work, the reviewer comments contained several valuable points regarding our interpretation and experimental design, which were addressed in the revised version of the manuscript. A point-by-point answer is appended below.

Reviewer #1 (Remarks to the Author):

This manuscript describes detailed structural and function work on a heterodimeric ABC exporter shifted to the outward-phasing state by a sybody. PELDOR/DEER data are of high quality and significance and clearly prove that this conformational shift happens. Several important conclusions about the transport and inhibition mechanism can be drawn from this data. Mutants show that extracellular gate closure is required for facilitating NBD dimer separation and thus transport cycling. All findings and the thereof drawn conclusions are convincing. Combining structural, spectroscopic and simulation results gives very interesting new insights into the function of this important class of heterodimeric membrane - therefore I recommend publication of this article in Nature Communication.

We thank reviewer#1 for this very positive assessment of our manuscript, in particular with regard to our DEER analyses.

Reviewer #2 (Remarks to the Author):

Hutter et al provide a structural and biophysical analysis of the archaeal, heterodimeric ABC transporter TM287/288. The authors present crystal structures of outward-open TM287/288, one with a synthetic single domain antibody (sybody) bound on the extracellular side and two with nanobodies bound to the nucleotide binding domains. The structures are in complex with a non-hydrolyzable nucleotide analogue, ATPS. The authors say that the sybody was essential for achieving high resolution of the outward-open structure (3.2 Å). They propose a mechanism in which the sybody allosterically inhibits the transporter and attempt to use it as a probe to show that closure of the extracellular gate is required for dissociation of the NBD dimer. They include some molecular dynamics analysis, mutagenesis data, and DEER and SPR measurements to attempt to support their mechanistic proposals.

We doubt the usefulness and impact of the study due to the following:

• The fundamental point / assumption of the study is that TM287/288, an archaeal 'model' system can provide valuable insight into eukaryotic ABC exporters including MRP1 or CFTR (lines 331-333). In lines 331-333 (part of discussion), we mentioned two studies on MRP1 and CFTR in which mutations in the extracellular gate were studied at the biochemical level. In our view, conformational coupling between extracellular gate and NBDs in analogy to our finding with TM287/288 may explain the results obtained in these studies. Of note, these three lines and the structural comparisons of outward-facing ABC exporter structures (lines 132 – 138) were the only instances in the manuscript where eukaryotic transporters are mentioned/discussed. To avoid misunderstandings regarding the value of our work for eukaryotic transporters, we have reworded lines 331-333 in our manuscript (lines 333-336 in revised version):

“We hope that our results provide a mechanistic framework to further study the functional role of the extracellular gate of type I ABC exporters and to investigate the molecular underpinning of disease-causing mutations found in the extracellular region of medically important ABC exporters such as MRP1 and CFTR[1, 2].”

In our view, this concept is invalid. First, although the authors assume that TM287/288 is a drug transporter, there is no evidence supporting this (see also below).

TM287/288 transports daunorubicin and BCECF-AM when expressed in *Lactococcus lactis*, as we reported in Hohl et al., NSMB, 2012.

Its role in transporting drugs in the native organism (*T. maritima*) has not been demonstrated. As a matter of fact, for many well-characterized drug transporters (e.g. P-glycoprotein or the yeast transporter Pdr5) the physiological substrates are unknown. The reason behind this “shortcoming” is the difficulty to identify physiological substrates, which are often hydrophobic and not easily accessible by metabolomics approaches. In our view, the identification of physiological substrates in the thermophilic bacterium *Thermotoga maritima* (optimal growth temperature at 80 °C) is extremely difficult or even impossible.

It is therefore very unlikely that insight gained from this prokaryotic ABC transporter of unknown physiological function can be applied to any of the eukaryotic systems.

First, we do not make the statement that our findings can be translated 1:1 to eukaryotic ABC transporters.

Second, we wish to emphasize that all type I ABC exporters share the same fold and can be superimposed at the structural level. Hence, there are mechanistic principles (such as for example the closure and opening of the highly conserved NBDs in response to ATP binding and hydrolysis) which are universally conserved in all ABC transporters.

Third, there are many mechanistic similarities between TM287/288 and the multidrug resistance eukaryotic Pgp, as can be appreciated based on DEER studies (see DEER data published in Verhalen et al., *Nature*. 2017 543(7647):738-741): i) AMPPNP does not shift the equilibrium to the OF state (in contrast to homodimeric exporters such as MsbA); ii) Vanadate-ADP-Mg is most effective in trapping the highest OF fraction in the ensemble; iii) The EtoQ mutation enhances the fraction of OF state in the presence of ATP-Mg; iv) Distance distributions in the extracellular region are broad and components similar to those of the apo state are present suggesting a heterogeneous transitions state.

Second, the authors themselves demonstrate that there is major heterogeneity in the degree of extracellular gate opening amongst ABC exporters (lines 281-291 and Fig. S6), exemplifying significant mechanistic differences.

We agree with the reviewer that there are different degrees of extracellular gate opening of ABC exporters, as structures and also DEER studies clearly show. However, there is one common principle that applies to most ABC exporters (exceptions are CFTR, which is an ATP-gated Cl-channel and PgIK which has a distinct transport mechanism): In order to release substrates to the periplasm/cell exterior, the extracellular gate must open and in order to revert to the inward-facing state, the extracellular gate must close again. While mechanistic details likely differ among ABC transporters, this basic mechanistic principle must be highly conserved.

If the authors want to test their hypotheses and draw conclusions concerning the mechanism of MRP1 or CFTR, they should study these proteins directly rather than relying on an unjustified model system.

It was not our intention to draw direct conclusions concerning the mechanism of MRP1 or CFTR (in fact, we mentioned these transporters as a side note in lines 331-333 in the discussion, which were rephrased/tamed in the revised manuscript). Rather, our intention was to elucidate conserved mechanistic principles of ABC exporters, namely the poorly understood re-setting step of the transport cycle, which involves the separation of the highly conserved NBD dimer after ATP hydrolysis. While performing pertinent experiments with MRP1 and CFTR go beyond the scope of this manuscript, we are confident that our findings will motivate other researchers to conduct experiments on MRP1 or CFTR and other ABC transporters. Importantly, our manuscript contains a validation of our findings with the bacterial ABC transporter EfrEF. Hence, our results are not limited to TM287/288.

- The authors report an outward-facing state of TM287/288, but given that this is not the first outward-open structure, it is unclear how this is relevant for the broader scientific audience.

TM287/288 is the third structure of a heterodimeric ABC transporter with asymmetric ATP binding sites (degenerate and consensus site) in an outward-facing state. The structure is nevertheless important, because it allows for a comparison with previously solved structures of TM287/288 representing the inward-facing state. It should also be mentioned that all structures of TM287/288 are of high resolutions, which is why this transporter had been used by several groups for molecular dynamics simulations (e.g. Göddeke et al., JACS, 2018).

Is it to understand substrate transport? If so which substrates are transported?

The outward-facing state is the low affinity state for transport substrates. Hence, structures of outward-facing ABC exporters are typically substrate-free, even if substrates are added (see also MRP1 structure, Johnson et al., 2018, Cell or ABCG2 structure, Manolaridis et al., Nature, 2018). Hence, the known substrates of TM287/288 (daunorubicin and BCECF-AM) are not expected to bind to outward-facing TM287/288.

In the abstract (lines 16 and 17) the authors say that for ABC exporters, 'extracellular gate opening and closure are key steps of the transport cycle', yet this concept is completely unproven.

Extracellular gate opening and closure happens during the transport cycle, this is in our view undoubted and can be seen in the various inward- and outward-facing structures. We are surprised to hear that this reviewer considers this as unproven.

We conclude that the study does not advance our mechanistic understanding of this ABC transporter (because we don't know what it is supposed to transport) nor of ABC transporters in general because there is no evidence supporting the idea that conclusions can be drawn for eukaryotic systems. It is a pity the authors did not dedicate their efforts to an ABC transporter of known function, where biophysical insight could be used to explain functional properties.

We disagree with these statements. In this study we investigated the coupling of the extracellular gate and the NBDs of a bacterial ABC exporter in the context of the re-setting step (conversion from outward-facing to inward-facing state). Importantly, the re-setting step occurs after substrate release, hence it represents the substrate-independent part of the transport cycle. As reviewer#2 may agree on, the closed NBD dimer – having two ATP sandwiched at its interface – is a highly conserved structural hallmark of outward-facing type I ABC exporters (and of ABC transporters in general). As a matter of fact, the closed NBD dimer with its large interaction interface needs to dissociate again – otherwise the transporter cannot revert back to its inward-facing state. This is a universal mechanistic “problem” that evolution had “solved” for all ABC transporters.

Using unique sybodies that specifically recognize the outward-facing state of TM287/288 as conformational probes, a previously uncharacterized extracellular gate mutant and high quality DEER analyses and functional/biochemical assays, we could show that extracellular gate closure is required for efficient opening of the closed NBD dimer after ATP hydrolysis. We are therefore convinced that our study touched on a very important and highly conserved mechanistic principle of ABC transporters, namely the opening of the closed NBD dimer in the re-setting step and its allosteric link to the extracellular gate. Importantly, extracellular gate mutants in EfrEF clearly show that this conformational connection between extracellular gate and NBDs exists in another bacterial transporter. As a side note, we also consider bacterial ABC transporters as important, not only eukaryotic ones.

We now clearly state in the revised manuscript that the extracellular gate aspartates are only conserved in bacterial, but not eukaryotic ABC exporters (line 176):

“Of note, these aspartates are conserved in bacterial ABC exporters (Fig. 4b), but not in eukaryotic members of the family.”

Having said this, the mechanistic problem of NBD dissociation also exists in eukaryotic ABC exporters and experimental clues that the NBDs are allosterically connected to the extracellular gate can be found in Weigl et al., *Mol Pharmacol*, 2018.

In addition to the major criticism above and while the study may have technical merits and may prompt similar studies of better-characterized ABC systems, there are unfortunately numerous issues with the interpretation (or over-interpretation) of data, as well as too much speculation. We include a point-by-point list of these concerns below:

- Line 19: In the abstract, the authors say that the sybody recognizes TM287/288 exclusively in the presence of ATP. On line 117 they say that the fully dimerized NBDs sandwich binds two ATP molecules. Actually, ATPs is bound - This is an important difference seeing as in a previous structure of TM287/288, and supported by subsequent DEER studies, the transporter remained in an inward-

facing state when one AMP-PNP molecule was bound to the degenerate catalytic site (see references below). How can the authors be sure that the structure with ATP bound will be the same as that in complex with ATP?

Reference: Crystal structure of a heterodimeric ABC transporter in its inward-facing conformation; Hohl et al, NSMB. 2012 March 25; 19(4):395-402. doi: 10.1038/nsmb.2267.

Reference: Exploring conformational equilibria of a heterodimeric ABC transporter; Timachi et al, eLife. 2017 Jan 4;6. pii: e20236. doi: 10.7554/eLife.20236.

We thank the reviewer for bringing up this valid point. Indeed, the presented outward-facing TM287/288 structure in complex with the sybody was obtained in the presence of ATPgS-Mg, not ATP-Mg. However, we also obtained the identical structure with crystals containing ATP-Mg (the crystals diffracted slightly worse (3.4 Å), this being the reason why we show the structure with ATPgS-Mg bound. We now mention the fact that we obtained the same structure with ATP in line 90:

“An identical structure at slightly lower resolution (3.4 Å) was also obtained in the presence of ATP-Mg (not shown). “

With regard to AMP-PNP, which was used to obtain the first inward-facing TM287/288 structure (Hohl et al, NSMB, 2012): We could show by DEER that AMP-PNP binding is not sufficient to switch the transporter to its outward-facing state (Timachi et al., eLife, 2017). In the same DEER paper, however, we could show that ATPgS can switch TM287/288 to its outward-facing state, in particular if it carries the E517Q mutation. The DEER study further revealed that the outward-facing states look identical if ATP or ATPgS are added.

Although AMP-PNP and ATPgS are considered as non-hydrolyzable ATP analogous, many researchers are not aware of the fact that ATPgS is cleaved by ATPases at a slow rate, whereas AMP-PNP is typically not cleaved at all. Hence, besides the exact chemistry, this is another important difference between these two nucleotide analogous.

- In the abstract and throughout the manuscript (lines 146, 178, 214, 286) the authors state that the sybody shifts the transporters conformational equilibrium to the outward-facing state. In Figs. 2c and 4f, they show that in the presence of ATP-EDTA, Sb_TM#35 promotes a more outward-facing conformation. However, in the absence of ATP the sybody does not bind. Is it simply a case that the sybody does not allow full closure of the extracellular side after it binds? If so then why would this inhibit ATP hydrolysis?

We used ATP-EDTA for the DEER analyses because under these conditions the transporter assumes both the inward- and the outward-facing state at the same time. Hence, the two states are close to equilibrium, which means that the system is sensitive to perturbations as manifested by strong equilibrium shifts towards the outward-facing state upon addition of the sybody and upon introduction of the extracellular gate mutation. The shift occurs because i) the sybody selectively binds to and thereby stabilizes the outward-facing state or ii) the extracellular gate mutant destabilizes the inward-facing state.

As we state in line 145, there are no steric clashes that would prevent binding of the sybody to the inward-facing transporter. Hence, subtle changes in the sybody epitope on the extracellular wing as the extracellular gate opens need to be responsible for the state-specificity of the sybody. We agree with the reviewer that sybody binding and extracellular gate closure are mutually exclusive. Hence, the sybody traps the transporter in its outward-facing state, which slows down transporter cycling

and hence ATPase activity (an analogy is the EtoQ mutation in the NBDs, which traps the transporter in its outward-facing state in a much more pronounced manner).

- Several state-specific sybodies were generated for TM287/288 before, including Sb_TM#35 (see reference below). In this previous study, it was demonstrated that another sybody, Sb_TM#26, binds to both wild type and TM287/288_E517A in the presence of ATP-Mg with a higher affinity than Sb_TM#35. Why did the authors choose Sb_TM#35 for structural studies?

Reference: Synthetic single domain antibodies for the conformational trapping of membrane proteins; Zimmermann et al, eLife. 2018 May 24; 7. pii: e34317. doi: 10.7554/eLife.34317.

It is correct that the sybody that was used in this study was isolated before (the respective citation – Zimmermann et al. – is found in line 88 of the initial and revised version of the manuscript). In Zimmermann et al., we identified more than 10 state-specific sybodies, which were all tested for crystal formation. With several of the sybodies, crystals were obtained, but only the crystals with Sb_TM#35 diffracted sufficiently well for structure determination.

Crystallization of membrane proteins is a cumbersome process, which cannot be fully rationalized. Finally, one has to screen many conditions/mutants/binders to reach the goal of solving a structure. In this case, it was not the best sybody in terms of affinity that resulted in a structure.

- Lines 70-71 and 275: The authors state that single domain antibodies were instrumental/ essential for solving the structure in an outward-facing state. We assume the authors attempted to trap an outward-facing state in the absence of a sybody or nanobody.

Indeed we repeatedly tried to obtain crystals of outward-facing TM287/288 in the absence of sybodies or nanobodies. Further, we tried around 20 – 30 different nanobodies (from alpacas) and sybodies. However, only with Sb_TM#35 we obtained crystals that diffracted sufficiently well for structure determination. That's why we state that this sybody was essential for solving the outward-facing TM287/288.

On line 194, and shown by molecular dynamics simulations in Fig. S11, the authors state that the outward-facing conformation is very stable in the absence of sybody. We appreciate that Fig. S1 shows that the sybody forms crystal contacts, but there are a number of ABC exporter structures in outward-facing conformations, determined at higher resolutions by either X-ray crystallography or cryo-EM, in the absence of conformational binders.

Indeed, outward-facing TM287/288_E517A in the presence of ATP-Mg or ATPγS-Mg is a stable conformation. Not only MD simulations show this, but also DEER analyses (Timachi et al., eLife,2017).

Sb-TM#35 played a dual role in the structure determination process: i) it further stabilized the outward-facing state, and in particular the opened extracellular gate which was shown to be rather flexible by DEER (Timachi et al., eLife,2017) and ii) it mediated crystal contacts as shown in Fig. S1. Our statements that a conformation-specific sybody was essential to obtain a high resolution crystal structure only apply to TM287/288 (see lines 70-71, and line 279).

We did not make the statement that such conformation-specific binders are essential tools for structure determination of outward-facing ABC transporters in general. But they are likely to be helpful in many other cases.

- Line 79: The authors acknowledge their previous work showing that a Walker B variant (TM287/288 E517Q) was almost completely trapped in an outward-facing state in the presence of ATP-Mg or ATPS-Mg (eLife reference below). The ATPase activity of this variant was reported to be extremely low (0.165 nmolPi min⁻¹ mg⁻¹, which was 0.19% of the wild type activity). Was the EtoQ variant not a valid structural target? Was it necessary to reduce the activity any further by introducing the EtoA mutation?

We initially tried to crystallize TM287/288_E517Q in the presence of ATP-Mg, but then realized that due to the high protein concentrations needed for crystallization, 5 mM of ATP is completely converted to ADP within less than 5 days, which was too short to obtain crystals. This was why we had introduced the E517A mutation, which exhibited a further decreased residual ATPase activity by a factor of around 6. However, it would have probably been possible to obtain crystals with the E517Q mutant in the presence of the slowly-hydrolyzed ATP analogue ATPγS. This was however not tested and we do not consider it as important to test this.

We appreciate that Fig.5b (and explained in lines 240-257) shows the t_{1/2} of the outward-facing state is generally longer for the EtoA variant, but these experiments were not done with ATPS (used for the structure). Why did the authors use ATPS-Mg? What is the t_{1/2} of the variants in the presence of ATPS-Mg? These points are important to address seeing as TM287/288 responds differently to different nucleotide analogues (see references below).

Reference: Exploring conformational equilibria of a heterodimeric ABC transporter; Timachi et al, eLife. 2017 Jan 4;6. pii: e20236. doi: 10.7554/eLife.20236.

Reference: Crystal structure of a heterodimeric ABC transporter in its inward-facing conformation; Hohl et al, NSMB. 2012 March 25; 19(4):395-402. doi: 10.1038/nsmb.2267.

As we outlined above, we obtained crystals of outward-facing TM287/288_E517A + Sb_TM#35 both in the presence of ATP-Mg and ATPγS-Mg. Because both of these nucleotides are hydrolyzed (however, at different rates) and behave very similarly in DEER (Timachi et al, eLife, 2018), we decided to investigate ATP-Mg (hydrolyzing condition) and ATP-EDTA (ATP binding only, no hydrolysis) in Fig.5. As a matter of fact, Fig. 5 has high information content with these two nucleotide conditions. Therefore, we feel that inclusion of ATPγS data adds more complexity without providing further mechanistic insights.

- The authors see very different effects in detergent and nanodisc environments. Fig. 2b shows that in detergent Sb_TM#35 inhibits ATPase activity by ~80%, at a concentration of 1 μM, and has an IC₅₀ of 66 nM. Fig. S2b shows that in nanodiscs Sb_TM#35 only inhibits by ~40% even though it was added at a much higher concentration (10 μM). Why was 10 μM Sb_TM#35 added in Fig. S2b? Is there still inhibition at the concentrations used in Fig. 2b?

The reviewer is right that TM287/288's ATPase activity cannot be inhibited as efficiently in nanodiscs by Sb_TM#35 as in detergent. If we add only 1 μM sybody, we did not observe significant inhibition. We explicitly note this now in the revised manuscript in lines 101-103:

“Indeed, the sybody inhibited the ATPase activity of TM287/288 in detergent (IC₅₀ of 66.1 nM, Fig. 2b) as well as reconstituted in nanodiscs (Fig. S2b). Of note, inhibition was less efficient in nanodiscs, presumably due to impaired epitope accessibility of the sybody in the membrane context.”

How do the raw ATPase activities (nmolPi min⁻¹ mg⁻¹) compare in detergent and nanodiscs? The ATPase activity in nanodiscs was around 10 times lower than in detergent, which is reminiscent of our previous findings in proteoliposomes (7 fold reduced ATPase activity, see also Hohl et al., NSMB, 2012). Also the Hoechst 33342 stimulation profile determined in nanodiscs is similar to the one measured in proteoliposomes. In essence, TM287/288 reconstituted in nanodiscs or proteoliposomes behaves very similarly.

Have the binders been tested in proteoliposomes? These comparisons are important seeing as it was previously reported that TM287/288 has ~7-fold lower ATPase activity when reconstituted in E.coli lipid/PC liposomes compared to detergent (see reference below)

Reference: Crystal structure of a heterodimeric ABC transporter in its inward-facing conformation; Hohl et al, NSMB. 2012 March 25; 19(4):395-402. doi: 10.1038/nsmb.2267.

The sybody binds to the extracellular wings, i.e. opposite of the NBDs. During the formation of proteoliposomes, the transporter reconstitutes in both directions, i.e. with NBDs in the liposome lumen or the NBDs exposed to the bulk solution. But only transporters oriented with NBDs pointing to the bulk solution contribute to ATPase activity. But exactly in this orientation, the sybody epitope is in the liposome lumen and thus not accessible. In other words, it does not make sense to perform the sybody inhibition experiment in proteoliposomes and this was the main reason we used nanodiscs, for which the orientation problem does not exist.

- The authors make a statement on line 100 that, ‘hence, the sybody addresses an epitope that is accessible by biopharmaceuticals in the cellular context.’ The authors need to explain what they mean by this statement, especially in light of the disparity between their detergent and nanodisc results and lack of assessment in proteoliposomes, as mentioned above.

We agree with the reviewer that this is an over-statement (owing to the fact that inhibition by sybody in nanodiscs is rather limited). We have thus deleted this statement in the revised manuscript.

- Lines 147, 176, 239, Figs. 2c, 4f, 5b: Why was ATP-EDTA used to create non-hydrolyzing conditions when ATPS was used for the structures?

As mentioned above, ATPgS is hydrolyzed by TM287/288, but there is absolutely no hydrolysis in the presence of ATP-EDTA (as shown in detail in Timachi et al., eLife, 2017).

- Line 150 and Fig. 2c: The authors state that sybody binding changes the distance between two spin labels positioned in the wing underneath the sybody. Do the authors think that this distance change has any mechanistic implications?

Yes, we think so and added the following sentence to line 155:

“This suggests that the sybody acts as a wedge at the opened extracellular wing.”

- Lines 141 and 165: The authors call the sybody a ‘molecular clamp’, but Fig. 2a does not show this clearly. A 90 rotation is also required to demonstrate how the sybody acts as a molecular clamp.

Fig. 2a shows a 90° rotation view already. We do not see the necessity to include the 180° and 270° view.

The authors could include the sybody in Fig. S4 (we assume the sybody has been removed here but this is not stated in the legend).

We thank the reviewer for spotting this. The legend of Fig. S4 now states that the sybody was removed.

- A number of TM287/288 variants are presented in this study for both structural and functional purposes. It would be beneficial to show SEC profiles of these mutants compared to the wild type protein and to make a comment on their stability.

We added the following sentence to the materials and methods section (line 384):

“Purified mutant proteins were all analyzed by SEC and did not differ in terms of elution profile and yield from the wildtype transporter.”

A big advantage of working with TM287/288 is its very high (thermal) stability. We have never observed any stability issues with mutants of this transporter.

We appreciate that the binding affinity of Sb_TM#35 and Nb_TM#1 is actually higher for the EtoA and 2xDtoA variants. However, the ATPase activities and percentage inhibition by Sb_TM#35 for the spin label variants is lower than for the wild type protein (Table S3).

Table S3 summarizes ATPase activities of spin-labelled DEER pairs of TM287/288 in the presence and absence of Sb_TM#35 (it shows only the new spin-label position at the extracellular gate of TM287/288 which were made as part of this study). It is indeed the case that some DEER pairs have a somewhat reduced ATPase activity compared to the wildtype transporter and the percentage inhibition with the sybody is lower. However, these measurements have nothing to do with ATPase activities of the 2xDtoA or EtoA variants.

- The authors use different variants for different structures. TM287/288 in complex with Sb_TM#35 is with an EtoA variant, while both nanobody complexes contain a 2xDtoA/ EtoA mutation. Can the authors explain why different variants were used? How can the authors be sure that it is the sybody, rather than the type of mutation, that explains the difference in resolution of each complex?

With the alpaca nanobodies, we did not obtain crystals containing outward-facing TM287/288 with TM287/288_EtoA alone, but only if we introduced the combined 2xDtoA/ EtoA mutations. By contrast, with the sybody we obtained crystals of the outward-facing transporter when it harbored the EtoA mutation alone as well as the combined 2xDtoA/ EtoA mutations. To minimally deviate from wildtype TM287/288, we then focused our crystal refinement to the transporter containing only the EtoA mutation, which resulted in the 3.2 Å structure.

- Lines 196-211 and Fig. 4d,e: Why did the authors only study membrane reconstituted EfrEF and not TM287/288?

Fig. 4d shows drug-stimulated ATPase activity of reconstituted EfrEF. Drug-stimulated ATPase activity of TM287/288 is shown in Fig. S2a.

The legend of Fig. 4e states: “Ethidium accumulation of *Lactococcus lactis* cells expressing wildtype EfrEF, the inactive Walker B mutant E515Q^{EfrF} or the extracellular gate mutants D41A^{EfrE} and D50A^{EfrF}”

or the corresponding double mutant (2xDtoA).”

Hence, in this case we looked at dye transport *in vivo* and not at a reconstituted transporter.

- Lines 258-262: Is it only data on TM287/288 for which these two major conclusions can be drawn? What about results from other ABC transporters?

Yes, these lines summarize the results of Figure 5. To our knowledge there is no other study that investigated the transition from the outward- to the inward-facing state in this manner, namely through the use of conformation-selective binders that probe the outward-facing state of an ABC transporter. Hence, this is a truly unique dataset for ABC transporters.

Given the fact that the closed NBD dimer and the extracellular gate are conserved structural hallmark of all outward-facing ABC transporters, it is likely that the conformational coupling we observed between extracellular gate and NBDs plays also a role in other ABC exporters. Future studies will hopefully provide answers to this question.

- Lines 265-267: As mentioned in the points above, we feel the choice of target was a poor one. The authors should use their state-specific antibody approach to explore the mechanisms of ABC transporters of known function. Have the authors explored the function of TM287/288? What is the effect of a TM287/288 knockout/knockdown?

We disagree with the reviewer in this point. TM287/288 had been extensively studied at the structural and biochemical level (including DEER studies). We also functionally characterized TM287/288 by expressing the transporter in *Lactococcus lactis* and could show that it transports daunorubicin and BCECF-AM (Hohl et al., NSMB, 2012). As outlined above, we consider it as technically very difficult if not impossible to study TM287/288 in its native hyperthermophilic organism *T. maritima*.

- Line 278: The authors say, ‘this means that ATP hydrolysis of only one nucleotide is sufficient to initiate dissociation of the NBDs’. Do they have any proof of this?

There is convincing biochemical evidence that even in ABC exporters containing two consensus ATP binding sites, only one ATP is hydrolyzed per catalytic cycle (e.g. Urbatsch et al., 1995, JBC; Orelle et al., 2003, JBC, Mittal et al., JBC, 2012). In heterodimeric ABC exporters containing asymmetric ATP binding sites, there is solid evidence that only the ATP molecule bound to the consensus site is hydrolyzed, because mutations introduced at the catalytic residues of the the degenerate only modulate, but do not abrogate ATP hydrolysis whereas the consensus site Walker B and Switch loop mutants are basically inactive (see also our work on the heterodimeric ABC exporter EfrCD: Hürlimann et al., FEBS Journal, 2017). We also have unpublished data at hand for TM287/288, which clearly show that mutations introduced at the catalytic residues of the degenerate do not strongly affect ATPase activity, whereas mutations in the respective residues of the consensus site basically abrogate ATPase activity (see also Fig. S7 in Bukowska et al., Biochemistry, 2015).

We nevertheless tamed our statement and now write in line 281 the following:

“This suggests that ATP hydrolysis of only one nucleotide is sufficient to initiate dissociation of the NBDs.”

- Lines 281-291 and Fig. S6: The authors compare structures of different ABC exporters and acknowledge major heterogeneity in the degree of extracellular gate opening. This exemplifies critical differences amongst ABC exporters and highlights how TM287/288 should not be used as a 'model' to understand other systems. Cross-talk between the extracellular gate and the ATPase cycle is not a general phenomenon of ABC transporters and may be specific to TM287/288.

We did not make the statement that our findings with TM287/288 can be translated 1:1 to other ABC transporters. However, we also introduced the extracellular 2xDtoA mutation into EfrEF and found it to react very similarly to TM287/288, in particular with regard to i) a decreased ATPase activity of the mutant and ii) abrogation of drug-induced ATPase activity (Fig. 4c and d, Fig. S2a). Hence, at least in the two tested systems the allosteric coupling between extracellular gate and NBDs exists.

As outlined above, extracellular gate opening/closure and NBD opening/closure are coupled events according to all currently available ABC exporter structure. Hence, there is no structure and also no functional evidence that an ABC transporter can exhibit dissociated NBD and an open extracellular gate at the same time. The differences among the compared structures in Fig. S6 relate to the degree of extracellular gate opening, which indeed differs among ABC exporters and which suggests differences regarding the details of the allosteric link between these two core elements. However, differences regarding the degree of extracellular gate opening do not exclude the likely possibility that there is a conformational link between extracellular gate and NBDs in most if not all ABC exporters.

- Lines 292-293: The authors say that using their nanobodies as conformational probes they were able to show that, 'extracellular gate closure is coupled to dissociation of the closed NBD dimer after substrate release and ATP hydrolysis'. We do not see how the nanobodies reveal this order of events (substrate release before extracellular gate closure). Surely this is an assumption as stated on line 326, 'it is reasonable to assume that the hydrolysis event happens after substrate release.'

We agree with the reviewer that we did not show/prove in our work that extracellular gate closure happens after substrate release (although this is a reasonable assumption). We have therefore deleted "substrate release" from this sentence in lines 292-293.

- Lines 307-311: Interesting mechanistic speculation, but may only be applicable to TM287/288. We feel that it is permitted to speculate about the mechanism in the discussion, as long as it is marked as speculation as we did.

- Lines 324-326: Is TM287/288 really a drug transporter? Showing ATPase stimulation by Hoechst is completely inadequate (Fig. S2) as many other unrelated transporters transport or interact with this compound.

Drug-stimulated ATPase activity is a widely accepted assay to study interactions of drugs with ABC transporters. We recently conducted a study on seven heterodimeric ABC exporters of *Enterococcus faecalis* that were all predicted to be multidrug transporters (Hürlimann et al., AAC, 2016). Our experiments revealed that only 3 out of the seven transporters were capable of drug efflux, including Hoechst 33342. Interestingly, we then conducted Hoechst-stimulated ATPase activity assays with the three identified drug transporters and also with one of the ABC transporter not

capable of drug transport (EF0942/41 in Fig. 5c: Hürlimann et al., AAC, 2016). Importantly, the ATPase activity of this non-drug transporter EF0942/41 was not stimulated by Hoechst, showing that this assay is specific. In Fig. 2c of the presented manuscript, ATPase activity is very strongly stimulated (as was also the case for TM287/288 in proteoliposomes: Hohl et al., NSMB, 2012). We interpret this as clear evidence that Hoechst 33342 is a substrate of TM287/288.

Before TM287/288 can be called a “drug transporter”, its role in transporting drugs in the native organism (*T. maritima*) must be proven. Perhaps it is not a drug transporter at all but a translocase of cell wall components or O-antigens?

As mentioned above, the identification of the physiological substrate of TM287/288 is experimentally very difficult if not even impossible.

- Lines 330-333: There is no evidence that the results in this study provide a mechanistic framework to understand other ABC exporters. Either test the hypotheses on MRP1 and CFTR directly or do not mention these transporters at all.

As part of this manuscript we could show that the conformational coupling between extracellular gate and NBDs exist for two bacterial ABC transporters, TM287/288 and EfrEF.

We rephrased/tamed the sentence in lines 330-333 (lines 333-336 in revised version) such that further studies are required for other (eukaryotic) ABC transporters:

“We hope that our results provide a mechanistic framework to further study the functional role of the extracellular gate of type I ABC exporters and to investigate the molecular underpinning of disease-causing mutations found in the extracellular region of medically important ABC exporters such as MRP1 and CFTR[1, 2].”

- Lines 383-393: For crystallization TM287/288 was first mixed with Sb_TM#35 before ATPS and MgCl₂ were added. The authors write that ATPS-Mg had to be pre-incubated with the complex for 5-6 days at 20 C before crystals were grown in order to obtain high resolution. Seeing as Sb_TM#35 only recognizes the outward-facing transporter, would it not have made more sense to first incubate TM287/288 with ATPS-Mg and then add Sb_TM#35?

In our view it does not matter whether ATPS-Mg or Sb_TM#35 is added first, because all components quickly equilibrate. We actually tested if changing the order of addition modified the DEER distances and we found no effects (data not shown). The reason why we incubated the sample for 5-6 days at 20°C was our serendipitous observation that crystals diffracted better if we did so. Why this was the case we don't actually know, but to be fully transparent, we included this information in the materials and methods section.

- Lines 394-404: Following on from the point above, the same is true for the complex with Nb_TM#1. Why was nanobody added before ATPS? In addition, why was the complex only incubated for 15 min prior to crystal growth when it is claimed above that a 5-6 day incubation was necessary for high resolution of the Sb_TM#35 complex? In the manuscript (lines 86 and 104) the authors say that the complex with Nb_TM#1 did not diffract well enough to build a reliable model (3.5 Å compared to 3.2 Å for Sb_TM#35). Is this simply due to the differences in sample preparation rather than any benefit of the sybody over the nanobody?

Crystals with Sb_TM#35 in general diffracted better than those obtained with Nb_TM#1, also with

short incubation times of the Sb_TM#35 complex. Crystallization is not a fully rational process. Crystal structures and their resolution, however, are speaking for themselves.

- Lines 383-415: Why were different concentrations of ATPS used for the different complexes (2.5 mM for Sb_TM#35 and Nb_TM#2, 5 mM for Nb_TM#1)? Why was a preparative SEC step used to remove excess nanobody but was not used to remove excess sybody?

Again, crystallization attempts are trial and error processes in which different conditions are tried out, including the addition of different nucleotide concentrations, the addition of sybodies/nanobodies before or after SEC to name a few. We simply reported in the most transparent way under which conditions the best diffracting crystals were obtained.

- Line 499: Why was ATP added for the alpaca immunizations yet ATPS was used for the structures? Why was the ATP:MgCl₂ ratio 3.3:1?

ATP-Mg was added to the protein in the cross-linking reaction in order to populate the outward-facing state. ATP-Mg was then washed away during subsequent purification steps and was not present anymore in the sample injected into the alpaca.

An excess of ATP over Mg was used because Mg is only a co-factor, whereas ATP is converted to ADP during the incubation.

- Lines 613-650: Why did the authors choose MSP1E3D1 (the authors write MSP1D1E3 but mean MSP1E3D1)? The authors state that, 'to spare membrane protein the ideal lipid:MSP ratio was determined beforehand by reconstituting empty nanodiscs,' (lines 638-639). Surely TM287/288 has to be included in the optimization procedure for the best lipid:MSP:TM287/288 ratio to be determined. SDS-PAGE and SEC profiles of the complexes should be provided to aid the explanations in this section.

We thank the reviewer for having spotted this error. Indeed, we used MSP1E3D1. We have corrected this in the revised version of the manuscript. In addition, we show SDS-PAGE and SEC profile of the complexes in new Fig. S2c, as requested by the reviewer.

- Fig. 1d: the figure legend is the wrong way around. SPR analyses in the absence of ATP is the upper panel and in the presence of ATP is the lower panel.

This error was corrected.

- Fig. 2b: can the authors explain why Nb_TM#1 only inhibits ATPase activity in detergent by ~40%? Seeing as it binds to the closed NBD dimer, would the authors not expect it to have a stronger inhibitory effect?

The binding affinity of Nb_TM#1 against wildtype TM287/288 in the presence of ATP-Mg is 184 nM (Table S2). When 1280 nM of Nb_TM#1 was added to TM287/288 for ATPase activity measurements (i.e. a concentration 7 fold above its KD), the inhibition was indeed around 40 %. Apparently the interaction of Nb_TM#1 with the closed NBD dimer is not strong enough to fully inhibit ATPase activity of TM287/288.

Reviewer #3 (Remarks to the Author):

The manuscript by Hutter et al reports on their study of bacterial ABC exporter TM287/288 using a novel synthetic single domain antibody that recognizes the transporter in the presence of ATP. They have been able to solve the crystal structure of the transporter in its OF state, which is significant since the OF structure of the transporter was not known before. They also use DEER, mutational studies, and molecular dynamics to support their main hypothesis that the "efficient extracellular gate closure is required to dissociate the NBD dimer after ATP hydrolysis".

The work is quite interesting in successful use of a synbody to capture the OF state of TM287/288, and in combining crystallography, DEER, biochemical, and computational data to build a comprehensive account of chemomechanical couplings in ABC exporters. The work certainly adds something to our knowledge of how ABC exporters work. Enough evidence has been provided for the main hypothesis of the paper; however, there are points that need to be addressed:

1) The generality of the finding. The authors should discuss their main finding within the context of ABC exporters in general. For instance, how the DEER data provided here compares to that reported for other ABC exporters such as MsbA, BmrCD, or P-gp.

The DEER data in this manuscript mainly report on the consequences of synbody binding and extracellular gate mutations on the conformational equilibrium of TM287/288. There are no studies on MsbA, BmrCD, or P-gp involving state-specific binders (nanobodies or classical antibodies), nor are there DEER studies on these transporters investigating mutations in the extracellular gate. This is why we cannot compare our DEER measurements directly with these transporters. Our previous paper (Timachi et al., eLife, 2017) contains extensive comparisons of general DEER analyses of TM287/288 with MsbA and BmrCD, but we feel that these findings do not need to be recapitulated in this manuscript.

Notably, as pointed out in the response to reviewer#2, DEER analyses revealed several mechanistic similarities between TM287/288 (Timachi et al., eLife, 2017) and the eukaryotic multidrug resistance transporter Pgp (Verhalen et al., Nature, 2017): i) AMPPNP does not shift the equilibrium to the OF state (in contrast to homodimeric ABC exporters such as MsbA); ii) Vanadate-ADP-Mg is most effective in trapping the highest OF fraction in the ensemble; iii) The EtoQ mutation enhances the fraction of OF state in the presence of ATP-Mg; iv) Distance distributions in the extracellular region are broad and components similar to those of the apo state are present suggesting a heterogeneous transitions state.

In terms of generality of the findings, we have conducted functional and biochemical experiments on the bacterial heterodimeric ABC exporter EfrEF (Fig. 4d and e). The findings for TM287/288 and EfrEF are highly similar.

2) The significance of the finding within the physiological context. The authors show that the synbody inhibits the ATPase activity by shifting the equilibrium towards the OF state. This is interesting but what does it mean in terms of the transport mechanism in the actual physiological context? This needs to be discussed.

We show that synbody as well as extracellular gate mutation perturb the conformational equilibrium of TM287/288 by shifting it towards the outward-facing state. As a consequence, ATPase activities

drop dramatically and drug stimulation of ATPase activity is lost. These perturbations shift the transporter away from its wildtype behavior, hence we learn only indirectly something about transporters physiology (i.e. ATPase activity of wildtype transporter is faster and can be stimulated by transport substrates). For the physiology of the transporter our findings indicate that coupling between extracellular gate and NBDs appears to be required for a proper functioning of the transporter in the presence of ATP (which in fact is present at mM concentrations in the cell).

3) The simulations were performed in a POPC bilayer. However, TM287/288 belongs to a gram-negative bacterium. I assume if one decides to simulate such a protein in a pure lipid bilayer, POPE would be the right choice. The authors should explain the reason for their choice and comment on how much they think this choice would have an effect on their simulation outcomes.

We agree with the reviewer that a POPE bilayer would have been a valuable option, because POPC is typically not contained in membranes of gram-negative bacteria. Having said this, it should be realized that for example *the E. coli* lipid bilayer only consists of around 60 % PE and in addition contains significant amounts of phosphatidylglycerols and cardiolipins. The lipid bilayer of hyperthermophilic organisms including *Thermotoga maritima*, the source organism of TM287/288, has yet a completely different lipid composition and contains quite unusual lipids [1].

Importantly, the outward-facing TM287/288 obtained from MD simulations after IF-to-OF conformational transition is structurally very similar to the outward-facing structure of the transporter determined by X-ray crystallography (Fig. S5). This shows that simulations in POPC bilayers lead to meaningful results, and, furthermore, renders it unlikely that the details of the lipid composition do not play a major role in the present case, at least not on the time scale covered by MD simulations. In fact, the influence of the lipid matrix has been studied in the context of another ABC transporter, SAV1866 [2,3]. Aittoniemi et al. [2] assessed the effect of the lipid environment (using POPC and DPPC) on SAV1866 and concluded “it seems very unlikely that the lipid environment would have any effect on the cytosolic regions of Sav1866 during the time covered in our simulations.” Furthermore, St-Pierre et al. [3] also studied SAV1866 but used a mixture of DLPE and DLPC lipids to mimic bacterial membranes. They compared their results to those of Aittoniemi et al. [1] and saw no effects due to the lipid matrix.

[1] Damsté JS, Rijpstra WI, Hopmans EC, Schouten S, Balk M, Stams AJ. Structural characterization of diabolic acid-based tetraester, tetraether and mixed ether/ester, membrane-spanning lipids of bacteria from the order Thermotogales. Arch Microbiol. 2007 Dec;188(6):629-41.)

[2] Aittoniemi, H. de Wet, F. M. Ashcroft, M. S. P. Sansom: A Simulation Study. PLoS Computational Biology. 6, e1000762 (2010).

[3] Jean-Francois St-Pierre, Alex Bunker, Mikko Karttunen and Normand Mousseau, Molecular Dynamics Simulations of the Bacterial ABC Transporter SAV1866 in the Closed Form. The Journal of Physical Chemistry B. 116, 2934–2942 (2012).

To clarify these points, we added the following sentences to the manuscript (line 662):

“Although lipid bilayers of gram-negative bacteria such as *E. coli* are mainly composed of phosphatidylethanolamines (PEs) and do not contain phosphatidylcholines (PCs), previous MD simulation studies on the ABC exporter Sav1866 suggested that lipid content of the bilayer does not

significantly influence the outcome of simulation experiments^{47,48}. Further, it should be noted that the source organism of TM287/288, the hyperthermophilic bacterium *Thermotoga maritima*, has a unique lipid composition⁴⁹, which is difficult to implement for MD simulations.”

4) Minor point: MD is mentioned by I can't find it being defined, which is supposed to be Molecular Dynamics.

“Molecular Dynamics (MD)” was added to line 61.

Reviewers' comments:

Reviewer #3 (Remarks to the Author):

Hutter et al have made a good-faith effort to address the concerns raised by myself and other reviewers in their revised manuscript. However, I disagree with their statement regarding the significance of the lipid composition.

The authors particularly refer to two simulations studies from 2010 and 2012 [Refs. 2 and 3 in their response letter] that effectively suggest no significant role for the lipid composition at least in the specific case of Sav1866. However, these studies use very short simulations of 40 ns (Ref. [2]) and 100 ns (Ref. [3]) to provide evidence for their claim. It is very likely that we do not see any significant differences between the PE- and PC-embedded Sav1866 or TM287/288 simulations within 40 or 100 ns. However, if we go beyond this time scale (e.g., 500 ns as in the current manuscript or potentially longer), the differences will start to show up. For instance, a recent study shows that Sav1866 simulated for a few microseconds behaves completely differently in different PE and PC environments (see Immadisetty et al ACS Cent. Sci., DOI: 10.1021/acscentsci.8b00480). Unfortunately, we cannot rely on short simulations, if we want to make statements about thermodynamic behavior of these transporters.

The authors also argue that since “the outward-facing TM287/288 obtained from MD simulations after IF-to-OF conformational transition is structurally very similar to the outward-facing structure of the transporter determined by X-ray crystallography”, the use of POPC lipids is justified. This argument was only valid if the X-ray crystallography was done in the physiological conditions, e.g., using the native lipids. However, the X-ray crystallography is done in a detergent environment. Therefore, it is likely that both the crystal structure and the simulations of TM287/288 are consistent with each other but not consistent with what we would have observed in a PE-containing lipid environment.

I believe the authors need to carefully examine these possibilities and discuss their results in this context.

Reviewer #4 (Remarks to the Author):

The authors have done a very good job to address the comments of the reviewers; not everything is presented in the paper and it would be useful to have the rebuttal available on line. In brief: (i) they have toned down the implications of their work for eukaryotic ABC exporters such as MRP1 and CFTR; (ii) they refer to papers that show that TM287/288 is a drug transporter (for most drug transporters the physiological substrates are unknown and perhaps do not exist); (iii) they have thoroughly addressed the various technical issues and choice of reagents as raised by reviewer 2.

The point on extracellular gate opening and closure could be discussed more elaborately.

The structure in the presence of ATP-Mg should be presented in the supplementary information rather than as data not shown.

With respect to the differences in inhibition by sy- and nanobodies, what are the K_m values of TM287/288 for ATP and other kinetic parameters when the protein is analyzed in detergent versus nanodiscs? The inhibition of the protein by Nb_TM#1, 40% inhibition irrespective of the concentration, is curious. Does the nanobody perhaps also affect the affinity for ATP or do the results reflect sample heterogeneity or differences in coupling (e.g. coupled activity affected but substrate-independent activity not affected or the other way around)?

The 10-fold higher ATPase activity in detergent presumably means that the ATPase is uncoupled from the translocator. It also implies that the ATPase activity is modulated by the presence of substrate when the protein is reconstituted in a lipid membrane (e.g. Fig. S2a) and not when it is present in detergent solution. Is this correct? The authors may want to elaborate a little more on this point.

POPC is often used for simulations, and although POPE membranes can be made in silico, vesicles/liposomes do not form from pure PE. As such mixtures of PE and PC together with an anionic lipid would be most relevant. The statement that the lipid content of the bilayer does not significantly influence the outcome of simulation experiments is worrying, because the activity of transporters is typically highly dependent on the lipid composition of the membrane and without PE or an anionic lipid most systems are inactive. The lipid independence questions the relevance of MD simulations.

We would like to thank the reviewers for their thoughtful comments. Our point-by-point answer is found below.

Reviewer #3 (Remarks to the Author):

Hutter et al have made a good-faith effort to address the concerns raised by myself and other reviewers in their revised manuscript. However, I disagree with their statement regarding the significance of the lipid composition.

The authors particularly refer to two simulations studies from 2010 and 2012 [Refs. 2 and 3 in their response letter] that effectively suggest no significant role for the lipid composition at least in the specific case of Sav1866. However, these studies use very short simulations of 40 ns (Ref. [2]) and 100 ns (Ref. [3]) to provide evidence for their claim. It is very likely that we do not see any significant differences between the PE- and PC-embedded Sav1866 or TM287/288 simulations within 40 or 100 ns. However, if we go beyond this time scale (e.g., 500 ns as in the current manuscript or potentially longer), the differences will start to show up. For instance, a recent study shows that Sav1866 simulated for a few microseconds behaves completely differently in different PE and PC environments (see Immadisetty et al ACS Cent. Sci., DOI: 10.1021/acscentsci.8b00480). Unfortunately, we cannot rely on short simulations, if we want to make statements about thermodynamic behavior of these transporters.

We agree with the reviewer that this is a potentially important issue that warrants close examination, and also with her/his statement about the limited relevance of short (40 or 100 ns) MD simulations. Therefore, we have carried out a set of ten additional MD simulations, each of length 500 ns, of the ATP-Mg bound wildtype X-ray structure embedded in a POPE (instead of a POPC) bilayer. The results (shown in the updated Fig. S11) confirm that the protein structure is very stable in the MD simulations and thermally fluctuates around an average structure that is very similar to the X-ray structure. So in sum, the results obtained in all three different environments (detergent in X-ray, POPC and POPE bilayers in MD) are consistent with each other.

Due to time restraints, we were not able to re-run the simulations from the IF to the OF conformation of TM287/288 embedded in POPE. Therefore, we cannot exclude that the corresponding trajectories may be different from the ones obtained in POPC, which are shown in Fig. S10.

To account for this possibility, we added some words of caution and cite Immadisetty et al in line 198:

“Although MD simulations and experimental data are in agreement, we cannot exclude different results if these rather long simulations were conducted in a bilayer containing other lipids such as for example POPE¹.”

In addition, one may also argue that the native lipid bilayer of *Thermotoga maritima* (the source organism of TM287/288) is very different from any simplified bilayer used for MD simulations. In fact, the exact lipid composition of this hyperthermophilic bacterium is unknown, but very unusual lipids have been reported (Damsté JS, Rijpstra WI, Hopmans EC, Schouten S, Balk M, Stams AJ. Structural characterization of diabolic acid-based tetraester, tetraether and mixed ether/ester, membrane-spanning lipids of bacteria from the order Thermotogales. Arch Microbiol. 2007 Dec;188(6):629-41.).

We hope that the reviewer understands that MD simulations in highly complex and near native lipid bilayers go beyond the scope of this manuscript.

The authors also argue that since “the outward-facing TM287/288 obtained from MD simulations after IF-to-OF conformational transition is structurally very similar to the outward-facing structure of the transporter determined by X-ray crystallography”, the use of POPC lipids is justified. This argument was only valid if the X-ray crystallography was done in the physiological conditions, e.g., using the native lipids. However, the X-ray crystallography is done in a detergent environment. Therefore, it is likely that both the crystal structure and the simulations of TM287/288 are consistent with each other but not consistent with what we would have observed in a PE-containing lipid environment.

Crystallographic analyses are never performed in true lipid bilayers (this also applies to lipidic cubic phase crystallization). A possible way forward is to determine cryo-EM structures of TM287/288 in nanodiscs containing native lipids. However, this is very complicated. A first challenge is to obtain enough biomass of *T. maritima* for lipid extraction, followed by tedious optimization of the nanodisc preparation with these lipids (as they are non-standard lipids) and finally the cryo-EM analysis itself. This would be an entire project on its own and goes far beyond the scope of this manuscript.

We would like to add that in a previous study, we performed DEER experiments on several spin-labeled pairs in the IF and OF states in detergent and in liposomes (Goeddecke et al., JACS, 2018). Liposomes were formed with polar *E. coli* lipids and egg phosphatidylcholine mixed at a weight ratio of 3:1. We could show that the distances are the same in the IF and OF state in detergent and liposomes for the NBDs and intracellular pairs. Interestingly, we observed that in liposomes the average distances between the extracellular spin pairs in the OF state slightly increased with respect to detergent, in line with the structure simulated in POPC bilayer (Goeddecke et al., JACS, 2018).

I believe the authors need to carefully examine these possibilities and discuss their results in this context.

We have modified the manuscript text (on p. 6 and p. 16) to discuss these aspects in the revised version.

Reviewer #4 (Remarks to the Author):

The authors have done a very good job to address the comments of the reviewers; not everything is presented in the paper and it would be useful to have the rebuttal available on line. In brief: (i) they have toned down the implications of their work for eukaryotic ABC exporters such as MRP1 and CFTR; (ii) they refer to papers that show that TM287/288 is a drug transporter (for most drug transporters the physiological substrates are unknown and perhaps do not exist); (iii) they have thoroughly addressed the various technical issues and choice of reagents as raised by reviewer 2.

The point on extracellular gate opening and closure could be discussed more elaborately.

In our opinion we elaborate on the point of extracellular opening and closure on various instances: i) in the context of other ABC exporter structure, ii) in the context of the 2xDtoA mutation and the equilibrium shift measured by DEER, iii) in the context of drug transport by EfrEF and iv) in the context of OF-IF conversion (i.e. the resetting step of the cycle). In addition, the extracellular gate is a prominent feature in our proposed model (Fig. 6).

An interesting aspect (which in the eyes of the reviewer may be somewhat lacking) is the role of the extracellular gate with regard to substrate transport. Because we do not provide data on this question in this manuscript, we can only speculate, and further studies will be required to dwell on the role of the extracellular gate in the context of substrate release.

The structure in the presence of ATP-Mg should be presented in the supplementary information rather than as data not shown.

We deposited the ATP-bound structure along with the other three structures, and show now the ATP-Mg-bound structure superimposed on the ATPyS-Mg structure in Fig. S4c. Further, we included the dataset in the crystallographic table (Table S1). With an RMSD of 0.21 Å, the ATPyS and ATP structures are highly similar also with respect to the nucleotides, therefore we decided that the structural analysis of the ATPyS-bound structure is sufficient. Of note, crystallographic data are slightly better for the ATPyS-bound structure due to less severe diffraction anisotropy, see also Fig. S13.

With respect to the differences in inhibition by sy- and nanobodies, what are the Km values of TM287/288 for ATP and other kinetic parameters when the protein is analyzed in detergent versus nanodiscs?

We thank the reviewer for asking this question, as it may be the root cause underlying the observed differences regarding the inhibition by sy- and nanobodies between detergent and nanodiscs.

To answer this question, we purified a fresh batch of TM287/288, kept an aliquot for measurements in detergent and reconstituted the remaining transporter protein in nanodiscs.

To be consistent with the sy- and nanobodies inhibition experiments, which were all conducted at 37 °C, we determined the kinetic parameters at 37 °C.

ATPase activity in DDM; $K_m = 7\mu M$, $v_{max} = 590 \text{ nmoles } P_i \text{ min}^{-1} \text{ mg}^{-1}$

ATPase activity in nanodiscs; $K_m = 11 \mu M$, $v_{max} = 18 \text{ nmoles } P_i \text{ min}^{-1} \text{ mg}^{-1}$

Apparent K_m values were basically identical, but the v_{max} for ATP hydrolysis was drastically lower in nanodiscs (around 30 fold, see also answer below).

Importantly, the sy- and nanobodies inhibition experiments were all conducted at 500 μM ATP, which is around 50-100 times above the apparent K_m . Therefore, the differences regarding the sy- and nanobodies in detergent vs. nanodiscs (in particular of Sb_TM#35) cannot be explained by different apparent K_m values for ATP, but are likely due to impaired epitope accessibility in nanodiscs.

We did not find an appropriate place in the manuscript to accommodate this data, as it is not of direct importance for the interpretation of our presented data.

The inhibition of the protein by Nb_TM#1, 40% inhibition irrespective of the concentration, is curious. Does the nanobody perhaps also affect the affinity for ATP or do the results reflect sample heterogeneity or differences in coupling (e.g. coupled activity affected but substrate-independent activity not affected or the other way around)?

We thank the reviewer for this valid question.

Concerning the 40% inhibition irrespective of the concentration of Nb_TM#2 (we clarified with the editor that reviewer#2 actually meant Nb_TM#2 and not Nb_TM#1), this was in fact an experimental limitation. In order to reliably measure ATPase activity of TM287/288 at 37 °C, the protein concentration needs to be at least 8 nM (as stated in line 486 in the materials and methods section). However, Nb_TM#2 has a very high affinity of around 1 nM (Table S2). Because inhibition data only make sense if $[TM287/288] < [Nb_TM\#2]$ (otherwise not all transporter molecules are occupied with a binder), the lowest binder concentration used in the assay was 20 nM (stated in line 488). Determination of an IC_{50} for Nb_TM#2 would have required ATPase activity measurements at much lower TM287/288 concentrations (i.e. 0.5 nM or even lower), which was, however, technically not possible.

We rephrased the text in the results in the following way (line 111):

“Nevertheless, this nanobody partially inhibits ATPase activity by around 30% already at the lowest assayed concentration of 20 nM (Fig. 2b). Because the TM287/288 concentration needed to be at least 8 nM to reliably measure ATPase activity, we could not determine the IC_{50} value for Nb_TM#2.”

To address the valid point whether addition of Nb_TM#2 affects the apparent K_m of TM287/288, we also determined v_{max} and K_m of TM287/288 in the presence of a saturating concentration (200 nM) of Nb_TM#2.

ATPase activity the absence of Nb_TM#2: $K_m = 7 \mu M$, $v_{max} = 590 \text{ nmoles } P_i \text{ min}^{-1} \text{ mg}^{-1}$

ATPase activity the presence of Nb_TM#2: $K_m = 5 \mu\text{M}$, $v_{\text{max}} = 388 \text{ nmoles } P_i \text{ min}^{-1} \text{ mg}^{-1}$

Apparent K_m values were basically identical ($7 \mu\text{M}$ in the absence and $5 \mu\text{M}$ in the presence of Nb_TM#2). In agreement with our IC_{50} determination for Nb_TM#2, v_{max} was reduced by 34 % in the presence of Nb_TM#2.

Hence, Nb_TM#2 does not change the apparent K_m of TM287/288 and only affects v_{max} . The mechanistic origin of v_{max} reduction by Nb_TM#2 therefore remains unclear. One could speculate that Nb_TM#2 binding slows down the conformational transitions of the transporter without affecting the thermodynamic equilibrium between the IF and OF state. The thermodynamic equilibrium is not affected because the affinity of Nb_TM#2 is identical in the absence and presence of ATP bound to TM287/288 (Table S2).

The 10-fold higher ATPase activity in detergent presumably means that the ATPase is uncoupled from the translocator.

The 10-fold difference for ATPase activity in detergent vs nanodiscs was an estimate we gave in our answer to reviewer#2 in the rebuttal letter based on measurements performed with different batches of TM287/288 on different days.

To determine the difference more accurately, we purified a fresh batch of TM287/288, kept an aliquot for measurements in detergent and reconstituted the remaining transporter in nanodiscs (see graphs above). These measurements showed that in nanodiscs the v_{max} of ATPase activity is reduced by a factor of 32.5 fold as compared to detergent. The apparent K_m values were similar ($11 \mu\text{M}$ in nanodiscs versus $7 \mu\text{M}$ in detergent).

There are in our view two scenarios which may explain these differences:

i) The ATPase is uncoupled from the translocator (as suggested by reviewer#4)

or

ii) There is a generally slower transition between the conformational states in nanodiscs owing to the lateral pressure of the membrane and/or the interaction of the transporter with lipids.

The first scenario is unlikely, because binding of Sb_#35 to the extracellular gate inhibits ATPase activity both in detergent and nanodiscs and likewise, the extracellular 2xDtoA mutant in the extracellular gate inhibits ATPase activity both in detergent and nanodiscs. This clearly shows that TMDs and NBDs are conformationally coupled in detergent solution (Fig. 2b, Fig. 4c, Fig. 4g).

Therefore, we propose that a decreased kinetics of the conformational cycle in a lipid environment causes the lower activity in nanodiscs.

It also implies that the ATPase activity is modulated by the presence of substrate when the protein is reconstituted in a lipid membrane (e.g. Fig. S2a) and not when it is present in detergent solution. Is this correct? The authors may want to elaborate a little more on this point.

To answer this question, we also determined the stimulation of ATPase by Hoechst 33342 in detergent.

As can be seen in the above graph, the ATPase activity of TM287/288 can be also stimulated in detergent. However, the maximal stimulation is only 1.8-fold in detergent solution vs. 10-fold in nanodiscs (Fig. S2a). It may be argued that in detergent solution, the detergent molecules may partially stimulate the ATPase activity already in the absence of added Hoechst 33342, which could explain the less pronounced ATPase stimulation with Hoechst 33342.

We first considered including this data into Fig. S2, but then realized that this will confuse readers, because all other measurements shown in Fig. S2 have been performed in nanodiscs. The reason we included Hoechst 33342 stimulation data in nanodiscs (Fig. S2a) was to show that drug stimulation is vanished in the 2xDtoA mutant. Due to the limited drug stimulation in detergent, we did not consider to carry out the measurements with the 2xDtoA mutant in detergent as well, because this does not add substantially to the paper.

POPC is often used for simulations, and although POPE membranes can be made in silico, vesicles/liposomes do not form from pure PE. As such mixtures of PE and PC together with an anionic lipid would be most relevant. The statement that the lipid content of the bilayer does not significantly influences the outcome of simulation experiments is worrying, because the activity of transporters is typically highly dependent on the lipid composition of the membrane and without PE or an anionic lipid most systems are inactive. The lipid independence questions the relevance of MD simulations.

This might be a possible misunderstanding. We did not intend to claim that the function of ABC transporters does not depend on lipids in general, and the reviewer is certainly right that there is a

wealth of published data showing that the ATPase activity of ABC transporters exhibits a dependence on lipids.

Due to technical reasons, it is challenging to perform MD simulations with mixed lipids and in the present case, one would have to build a lipid bilayer as it is present in *Thermotoga maritima*, the source organism of TM287/288. However, to the best of our knowledge the exact lipid composition of *T. maritima* is unknown, and it was reported that its bilayer contains a large number of unusual lipids (Damsté JS, Rijpstra WI, Hopmans EC, Schouten S, Balk M, Stams AJ. Structural characterization of diabolic acid-based tetraester, tetraether and mixed ether/ester, membrane-spanning lipids of bacteria from the order Thermotogales. Arch Microbiol. 2007 Dec;188(6):629-41.). Hence, simulations under conditions as proposed by reviewer#4 goes beyond the scope of this work.

To address the potential impact of lipids on MD simulation with TM287/288, we have carried out a set of ten additional MD simulations, each of length 500 ns, of the ATP-Mg bound wildtype X-ray structure embedded in a POPE (instead of a POPC) bilayer. The results (shown in the updated Fig. S11) confirm that the protein structure is very stable in the MD simulations and thermally fluctuates around an average structure that is very similar to the X-ray structure. So in sum, the results obtained in all three different environments (detergent in X-ray, POPC and POPE bilayers in MD) are consistent with each other.

Due to time restraints, we were not able to re-run the simulations for the IF-to-OF transition embedded in POPE. Therefore, we cannot exclude that the corresponding trajectories may be different from the ones obtained in POPC. We therefore state in line 198:

“Although MD simulations and experimental data are in agreement, we cannot exclude different results if these rather long simulations were conducted in a bilayer containing other lipids such as for example POPE¹.”

Despite of these disclaimers, we would like to emphasize that the relevance of the MD simulations presented in this work is, in our view, most clearly demonstrated in Fig. S5, which shows that the simulations initiated in the inward-facing conformation correctly predicted the outward-facing X-ray structure after undergoing a spontaneous large-scale conformational transition, although the MD simulations did not draw on any experimental information other than the starting (inward-facing) structure.

While this adds to the relevance of our findings, it cannot be excluded that the same simulations could have led to a different outcome with a POPE or a mixed lipid bilayer, nor can it be excluded that TM287/288 adopts a different conformation if the structure were determined by cryo-EM in a lipid environment in nanodiscs (see also our replies to reviewer#3 above).

We would like to close by saying that the main findings of our paper regarding the sy- and nanobodies, the extracellular gate mutant and the shifts of the conformational equilibrium do not critically depend on the results obtained by MD simulation. Rather, we consider the MD simulations as an interesting add-on, which nicely agree with our experimental findings.

REVIEWERS' COMMENTS:

Reviewer #3 (Remarks to the Author):

Authors have now addressed all of my concerns.

Reviewer #4 (Remarks to the Author):

I am satisfied with the rebuttal and revised manuscript. The authors have done a substantial amount of work to address the remaining questions, and their answers are generally convincing. They prefer not to include some of additional data in the manuscript, although the information could be presented in the supplemental information; I leave it to the editor to decide whether this is desirable or not.

My compliments to the authors of the paper.